# The RNA polymerase II subunit RPB-9 recruits the integrator complex to terminate *Caenorhabditis elegans* piRNA transcription

Ahmet C Berkyurek[1,2,†] iD, Giulia Furlan[1,2,†] iD, Lisa Lampersberger[1,2,†] iD, Toni Beltran[3,4] iD, Eva-Maria Weick[1,‡] iD, Emily Nischwitz[5], Isabela Cunha Navarro[1,2] iD, Fabian Braukmann[1,2] iD, Alper Akay[1,2,§] iD, Jonathan Price[1,2], Falk Butter[5] iD, Peter Sarkies[3,4] iD & Eric A Miska[1,2,6,*] iD

## Abstract

PIWI-interacting RNAs (piRNAs) are genome-encoded small RNAs that regulate germ cell development and maintain germline integrity in many animals. Mature piRNAs engage Piwi Argonaute proteins to silence complementary transcripts, including transposable elements and endogenous genes. piRNA biogenesis mechanisms are diverse and remain poorly understood. Here, we identify the RNA polymerase II (RNA Pol II) core subunit RPB-9 as required for piRNA-mediated silencing in the nematode *Caenorhabditis elegans*. We show that *rpb-9* initiates heritable piRNA-mediated gene silencing at two DNA transposon families and at a subset of somatic genes in the germline. We provide genetic and biochemical evidence that RPB-9 is required for piRNA biogenesis by recruiting the Integrator complex at piRNA genes, hence promoting transcriptional termination. We conclude that, as a part of its rapid evolution, the piRNA pathway has co-opted an ancient machinery for high-fidelity transcription.

**Keywords** integrator; piRNAs; RNA polymerase II; *rpb-9*; transcription termination

**Subject Categories** Chromatin, Transcription & Genomics; RNA Biology

**The EMBO Journal (2021) 40: e105565**

See also: **T Beltran** *et al* (March 2021)

## Introduction

The PIWI-interacting RNA (piRNA) pathway protects animal germlines from transposable elements and other selfish genetic parasites, thereby ensuring transgenerational genomic integrity and fertility (Bagijn *et al*, 2012; Lee *et al*, 2012). The piRNA pathway is widespread in metazoans and relies on the cooperation between small RNAs and Argonaute (AGO) proteins to silence targets both post- and co-transcriptionally; the precise molecular mechanisms through which it operates, however, vary between species.

*C. elegans* piRNAs play a fundamental role in recognizing non-self DNA elements, inducing their repression and promoting a multi-generational epigenetic memory of their silencing (Ashe *et al*, 2012; Bagijn *et al*, 2012; Shirayama *et al*, 2012; Lee *et al*, 2012). Moreover, they have been shown to regulate the expression of a subset of endogenous genes in the germline (Rojas-Ríos & Simonelig, 2018). Most *C. elegans* piRNAs are encoded by two specific clusters on chromosome IV, which contain thousands of intergenic and intronic individual piRNA transcription units enriched in Ruby motif-containing promoters (motif-dependent or type I piRNAs) (Ruby *et al*, 2006; Cecere *et al*, 2012; Gu *et al*, 2012; Billi *et al*, 2013). A smaller number of piRNAs are produced from other loci in the genome (Batista *et al*, 2008) and transcribed bidirectionally from transcription start sites (TSSs) flanked by YRNT motifs (motif-independent or type II piRNAs) (Ruby *et al*, 2006; Gu *et al*, 2012).

Mature *C. elegans* piRNAs are 21 nt small RNAs with a 5′ uracil bias (hence called 21U-RNAs), a 5′ monophosphate, and a 3′ hydroxyl group at their extremities, and are post-transcriptionally 2′-O methylated at the 3′ end (Kamminga *et al*, 2010; Montgomery *et al*, 2012). They bind to the PIWI subfamily proteins of the AGO family and guide them to complementary target transcripts.

Two PIWI proteins with germline-restricted expression, PRG-1 and PRG-2, have been identified in *C. elegans*, although only PRG-1 is required for maintaining wild-type piRNA populations (Batista *et al*, 2008; Das *et al*, 2008; Wang & Reinke, 2008). Unlike in other

1  Wellcome Trust/Cancer Research UK Gurdon Institute, University of Cambridge, Cambridge, UK
2  Department of Genetics, University of Cambridge, Cambridge, UK
3  MRC London Institute of Medical Sciences, London, UK
4  Institute of Clinical Sciences, Imperial College London, London, UK
5  Quantitative Proteomics, Institute of Molecular Biology, Mainz, Germany
6  Wellcome Sanger Institute, Wellcome Trust Genome Campus, Cambridge, UK
   *Corresponding author. Tel: +44 1223 334088; E-mail: eam29@cam.ac.uk
   †These authors contributed equally to this work
   ‡Present address: Structural Biology Program, Sloan Kettering Institute, Memorial Sloan Kettering Cancer Center, New York, NY, USA
   §Present address: School of Biological Sciences, University of East Anglia, Norwich, Norfolk, UK

animals, where PIWI/piRNA complexes silence their targets via a PIWI endonuclease "slicing" activity, PRG-1/piRNA complexes in *C. elegans* trigger target degradation indirectly. The imperfect base pairing of the piRNA with its complementary transcripts elicits a localized silencing amplification response in which the targeted recruitment of the RNA-dependent RNA polymerases (RdRPs) RRF-1 and EGO-1 leads to the generation of a secondary class of small RNAs (termed secondary endogenous siRNAs (endo-siRNAs) or 22G siRNAs) that are 22 nt long, have a 5′ guanine bias, and carry a 5′ triphosphate. These abundant molecules in turn promote robust and sustained silencing upon loading onto worm-specific AGO proteins (WAGOs) (Batista *et al*, 2008; Gu *et al*, 2009; Bagijn *et al*, 2012; Lee *et al*, 2012; Shirayama *et al*, 2012; Ashe *et al*, 2012; Mao *et al*, 2015).

Interestingly, this silencing response can be inherited transgenerationally and persist for several generations, even in *prg-1* mutant animals (Ashe *et al*, 2012). Once established, this long-term silencing is independent of the initial piRNA trigger and instead relies on the germline nuclear RNAi pathway, including the AGO protein HRDE-1 (Ashe *et al*, 2012; Buckley *et al*, 2012), the nuclear RNAi-defective proteins NRDE-1/2-4 (Ashe *et al*, 2012), the nuclear RNA helicase EMB-4 (Akay *et al*, 2017), and several chromatin-associated factors (Ashe *et al*, 2012; McMurchy *et al*, 2017). Together, the piRNA and the germline nuclear RNAi pathways establish and maintain transgenerational silencing via the inheritance of small RNA populations and the deposition of chromatin silencing marks in the germline (Ashe *et al*, 2012; Buckley *et al*, 2012; Lee *et al*, 2012).

Since PRG-1/piRNA complexes tolerate several mismatches when selecting for complementary transcripts, they can theoretically target any sequence (Bagijn *et al*, 2012; Lee *et al*, 2012; Shen *et al*, 2018; Zhang *et al*, 2018), including endogenous genes that are required to be expressed in the germline. To prevent silencing of germline-specific genes, some nematodes have evolved a protection mechanism involving the CSR-1/22G siRNA licensing pathway (Seth *et al*, 2013; Shen *et al*, 2018; Zhang *et al*, 2018), in which endogenous secondary 22G siRNAs associate with the AGO CSR-1 and prevent target recognition by the piRNA pathway.

Given the importance of piRNAs in the maintenance of genome stability, it is essential to understand how they are generated. *Caenorhabditis elegans* piRNAs are transcribed by the RNA Pol II holoenzyme (Gu *et al*, 2012); (Billi *et al*, 2013), which transcribes a range of coding and non-coding genes through three precisely defined and controlled steps: initiation, elongation, and termination (Adelman & Lis, 2012; Zhou *et al*, 2012). In eukaryotes, RNA Pol II is typically composed of 12 subunits (RPB1-12), of which 10 constitute the catalytic core, and two (RPB4 and RPB7 in yeast) are required for transcription initiation (Cramer, 2004). RPB1/AMA-1, the largest subunit, is essential for polymerase activity through its carboxy terminal domain (CTD) and, in combination with RPB9, forms the DNA-binding groove of the holoenzyme (Acker *et al*, 1997). RPB2, the second largest subunit, forms a cleft in which the DNA template and the nascent RNA transcript are kept in proximity (Acker *et al*, 1992; Ponicsan *et al*, 2013), while RPB6 is part of a structure that stabilizes the active enzyme on the DNA template (Acker *et al*, 1994; del Río-Portilla *et al*, 1999; Wani *et al*, 2014). The structural role of the other subunits is less well-documented (Sainsbury *et al*, 2015).

Faithful transcription is ensured by a proofreading mechanism called backtracking, during which the polymerase moves backward on the DNA template to correct base mis-incorporation events (Bondarenko *et al*, 2006; Churchman & Weissman, 2011; James *et al*, 2017). As a result, the 3′ end of the nascent transcript is displaced from the enzyme's active site, and the polymerase becomes transcriptionally inactive (Nudler *et al*, 1997; Komissarova & Kashlev, 1997; Thomas *et al*, 1998; Kettenberger *et al*, 2003; Wang *et al*, 2009; Cheung & Cramer, 2011). Backtrack recovery is required in order for the enzyme to resume transcriptional elongation, and often depends on cleavage of the backtracked RNA to generate a new 3′ end, which allows realignment with the active site (Chedin *et al*, 1998; Kuhn *et al*, 2007; Walmacq *et al*, 2009). Eukaryotic RNA Pol II has a weak intrinsic cleavage activity, which is strongly enhanced by the transcription elongation factor TFIIS (Izban & Luse, 1992).

RNA Pol II backtracking is associated not only with promoter-proximal pausing rescue and rapid transcription elongation, but also with transcription termination (Sheridan *et al*, 2019). Transcription termination is critical to determine borders between genes and to avoid interference with downstream-positioned loci, and it may be even more important for an organism like *C. elegans*, with a compact genome (Caenorhabditis elegans Sequencing Consortium, 1998). At nonpolyadenylated-type loci, such as those encoding for small nuclear RNAs (snRNAs), termination depends on specialized processing of the 3′ ends and often requires the Integrator complex, whose nuclease activity at specific cleavage sites is necessary for precursor transcript maturation (Uguen & Murphy, 2003; Baillat *et al*, 2005; Ezzeddine *et al*, 2011; Ezzeddine *et al*, 2012). Interestingly, transcription at motif-dependent piRNA loci shares evolutionary similarities to snRNA transcription (Beltran *et al*, 2019).

Here, by using a combination of genetics and biochemical approaches, we show that the RNA Pol II subunit RPB-9 is required to promote the Integrator-dependent cleavage of 3′ ends of nascent transcripts upon RNA Pol II backtracking for transcription termination at motif-dependent piRNA loci in *C. elegans*. In *rpb-9* mutants, a defect in transcription termination leads to a reduction in mature piRNA levels, which in turn results in a drastic depletion of HRDE-1-associated secondary 22G siRNAs and, ultimately, in the desilencing of two families of DNA transposons and a subset of somatic genes, likely in the germline.

## Results

### *rpb-9* is required for piRNA pathway integrity

In order to identify components of the piRNA pathway, we have previously described a forward genetics screen for animals defective in their ability to silence the "piRNA sensor", a germline-specific transgene containing a *gfp* reporter and responsive to the endogenous piRNA 21UR-1 (*pmex-5::eGFP::his-58::as21UR-1::tbb-2*) (Bagijn *et al*, 2012; Ashe *et al*, 2012; Weick *et al*, 2014) (Fig 1A). Wild-type animals efficiently repress this *gfp* transgene via the piRNA pathway, while mutants of piRNA pathway components, such as *prg-1* and *mutator* class genes, fail to do so and strongly desilence the piRNA sensor (Phillips *et al*, 2012). One of 22 independent mutants from our screen defined a new allele (*mj261*) of the gene *rpb-9*, which encodes for a conserved subunit of the DNA-dependent RNA

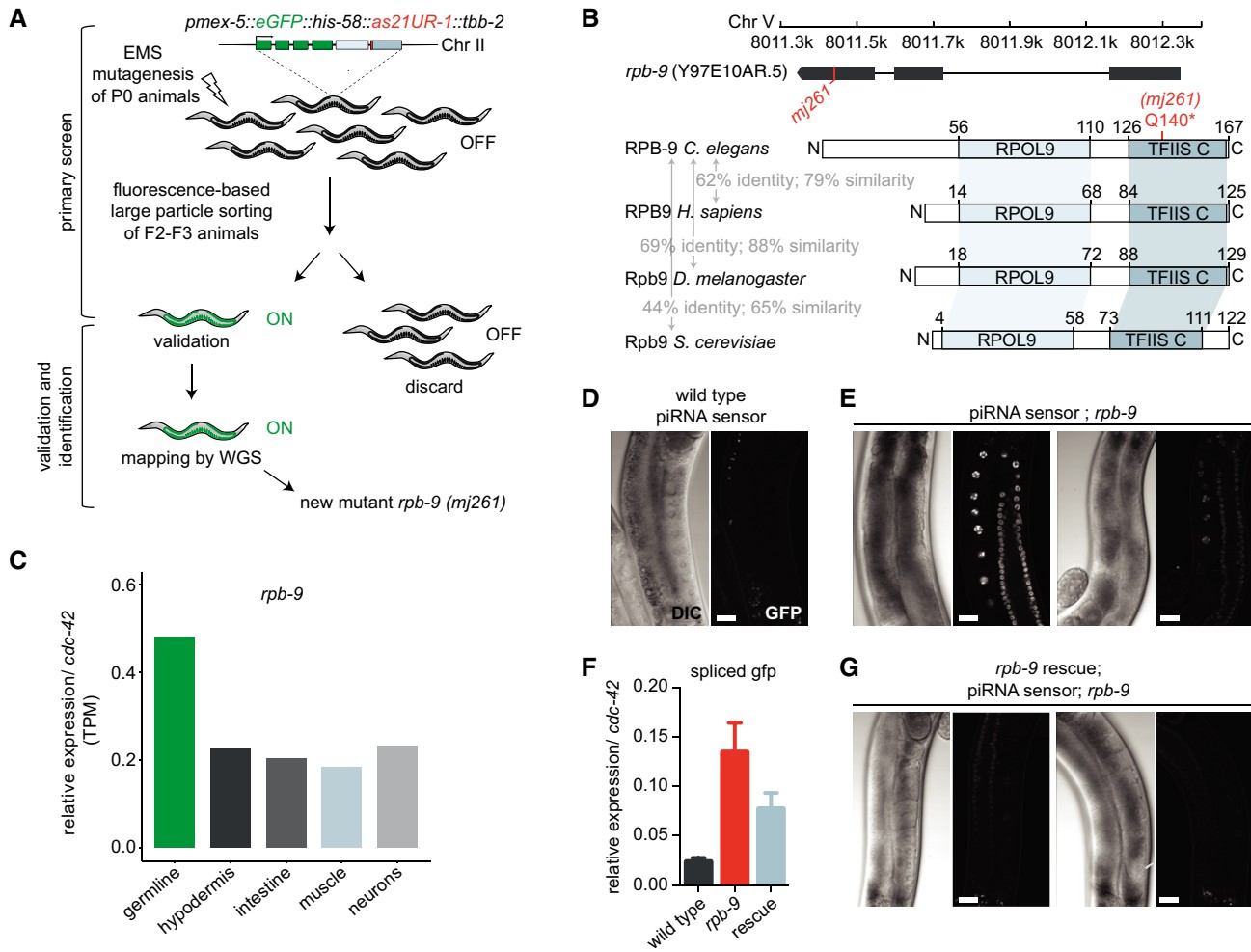

**Figure 1. *rpb-9* is required for piRNA pathway integrity.**

A   Schematic of the EMS mutagenesis screen carried out using the piRNA sensor and identification of a novel allele *(mj261)* of the *rpb-9* gene.
B   *rpb-9* locus, in scale (top) and comparisons of RPB-9 protein domains to homologs in other species (bottom). The introduced mutation is highlighted in red.
C   Expression patterns of *rpb-9* in adult animals by RNA sequencing.
D–G  Quantification of piRNA sensor expression. Representative DIC and fluorescence microscopy images of piRNA sensor expression in wild type *(mjIs144)* (D) and *rpb-9 (mj261) animals* (two images with varying GFP expression are shown, scale bar = 20 μm) (E), validation of transgene expression by RT–qPCR (*n* = 3, error bars represent SD) (F), and representative DIC and fluorescence microscopy images of piRNA sensor expression in the *rpb-9* rescue *(mj261; mjSi70)* animals (two images with varying GFP expression are shown). Scale bar = 20 μm (G).

Pol II enzyme (Cramer *et al*, 2000). Studies in yeast have proposed a role for Rpb9 in various steps of transcription, including start site selection and transcriptional elongation, and in transcription-coupled repair (Furter-Graves *et al*, 1994; Hemming *et al*, 2000; Li & Smerdon, 2002; Nesser *et al*, 2006). We decided to explore the possibility that *rpb-9* could provide a direct link between transcription and silencing in the context of piRNA-mediated repression in *C. elegans*.

*Caenorhabditis elegans* RPB-9 is composed of 167 amino acids. It comprises a central RPOL9 domain (a zinc ribbon) and a C-terminal zinc-finger motif-containing domain also found in the C-terminal portion of the transcription elongation factor TFIIS, hence named "TFIIS C". The identified point mutation introduces a premature amber stop codon at glutamine 140 (CAG to TAG mutation), within the core of the TFIIS C domain (Fig 1B). As expected, *rpb-9* mutants

do not express full-length RPB-9 protein at levels detectable via Western blotting, although we cannot exclude residual protein due to transcriptional read-through (Fig EV1A).

To explore the conservation of RPB9, we aligned the amino acid sequence of the *C. elegans* RPB-9 with its homologs in *Saccharomyces cerevisiae*, *Drosophila melanogaster,* and *Homo sapiens*. We observed that both the RPOL9 and the TFIIS C domains are conserved in all these species (with sequence identities from 44 to 69%) and that *C. elegans* RPB-9 also possesses an additional N-terminal portion, which does not contain any known or conserved domains (Figs 1B and EV1B). We also compared the *C. elegans* RPB-9 and TFIIS sequences, and found that their respective C-terminal domains share 33% sequence identity and 53% sequence similarity, suggesting they might perform similar functions (Fig EV1C).

In *C. elegans*, *rpb-9* is expressed both in the soma and in the germline, consistent with its role as RNA Pol II subunit (Serizay *et al*, 2020) (Fig 1C). When visualized under a fluorescence microscope, all *rpb-9* animals desilence the piRNA sensor in the germline, albeit at a variable level (Fig 1D (wild type), compare with Fig 1E (*rpb-9* mutant)). We measured the amounts of spliced *gfp* mRNA in these animals and observed increased levels of *gfp* expression, suggesting that *rpb-9* affects its targets prior to translation (Fig 1F).

To test whether the *rpb-9* mutation was causative for sensor desilencing, we constructed a synthetic transgene harboring the *rpb-9* coding sequence under control of a germline-specific promoter (*pmex5::rpb9::par-5*, *mjSi70*) and integrated it on chromosome I via the MosSCI technology (Frøkjaer-Jensen *et al*, 2008). This germline-specific allele restores wild-type levels of RPB-9 protein, as detected by Western blotting (Fig EV1A), and rescues the desilencing phenotype of *rpb-9* mutants (Fig 1F and G). Importantly, a similar transgene containing the Q140STOP mutation does not rescue the phenotype (Fig EV1D). All together, these data demonstrate that *rpb-9* is important for the robust function of the piRNA pathway.

## *rpb-9* is required for establishing piRNA-mediated silencing

The piRNA pathway involves multiple steps and intersects with other small RNA pathways. In order to understand whether *rpb-9* is required for transgenerational maintenance of silencing, we crossed *rpb-9* animals with animals expressing a germline *gfp::h2b* reporter (*ppie-1::eGFP::his-58::pie-1*) (Ashe *et al*, 2012; Buckley *et al*, 2012) and we assessed the transcriptional status of this transgene across generations upon a time-limited exposure to *gfp* RNAi (Fig 2A).

In wild-type animals, exogenous double-stranded (ds) *gfp* RNA uptake for one generation is sufficient to establish and maintain reporter silencing for several generations (Ashe *et al*, 2012; Buckley *et al*, 2012). Conversely, mutants of components of the multi-generational germline nuclear RNAi pathway, albeit able to efficiently establish silencing in sustained presence of ds *gfp* RNA, fail to maintain it once this trigger is removed. This is the case for *hrde-1,* previously identified as the major germline-specific AGO that binds secondary endogenous 22G siRNAs and directs initiation of the transgenerational germline nuclear RNAi pathway (Fig 2B and C) (Ashe *et al*, 2012; Buckley *et al*, 2012). Like wild-type animals, *rpb-9* mutants can efficiently (100%) initiate silencing upon *gfp* RNAi and are able to maintain it (91–100%) even when the initial stimulus is removed (Fig 2B and C, Appendix Fig S1). This suggests that *rpb-9* is not required for exogenous RNAi and its inheritance. Instead, it is required specifically for the endogenous piRNA-mediated silencing. Similarly to *rpb-9* mutants, *prg-1; prg-2* double mutant animals are efficient in both the establishment and the maintenance of exogenous dsRNA-induced silencing (Fig 2B and C, Appendix Fig S1). Hence, we conclude that *rpb-9* is required for piRNA-dependent transgene-silencing upstream of *hrde-1* and the multi-generational germline nuclear RNAi pathway, independently of the exogenous RNAi pathway.

## *rpb-9* is required to repress two DNA transposon families and a subset of somatic genes

In order to explore the genome-wide consequences of piRNA pathway disruption in *rpb-9* mutants, we performed a genome-wide

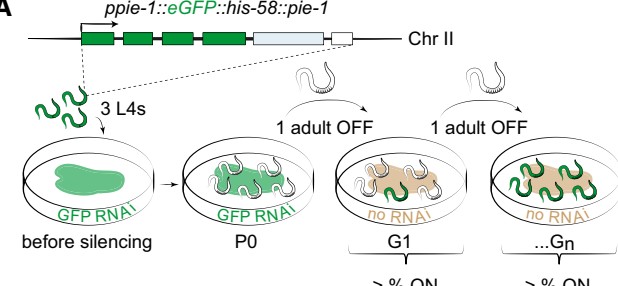

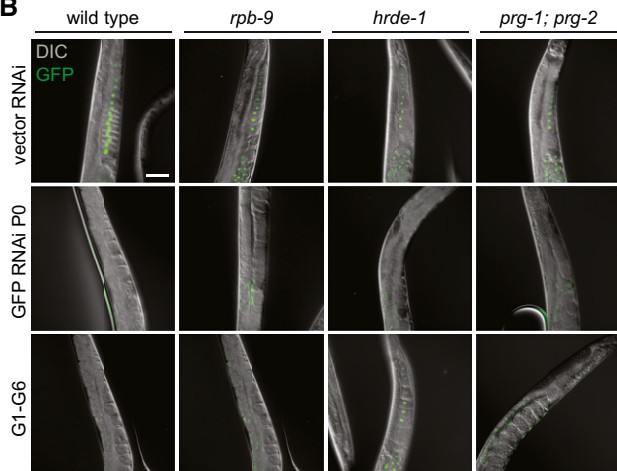

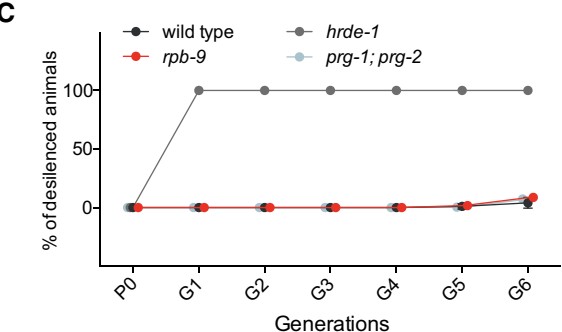

**Figure 2. *rpb-9* is required for the initiation step of the piRNA pathway.**

A Schematic of the germline-specific *pie-1::gfp::h2b::pie-1* sensor (top) and schematic of the experiment (bottom).
B Representative fluorescence images of transgene expression in wild-type, *rpb-9 (mj261)*, *hrde-1 (tm1200)*, and *prg1 (n4357); prg-2 (n4358)* animals in the parental (P0) and inheriting (G1-G6) generations. Scale bar = 50 μm.
C Quantification of the percentage of desilenced individuals in wild-type, *rpb-9 (mj261)*, *prg-1 (n4357); prg-2 (n4358)* and *hrde-1 (tm1200)* populations across generations.

transcriptome analysis (polyA-selected RNA-seq and ribosomal RNA-depleted RNA-seq).

We first analyzed the expression of DNA transposons and retrotransposons, both known to be targeted by the piRNA pathway. While no overall major changes were detectable in *rpb-9* mutants compared with wild type, we observed two de-repression events, which affected two independent autonomous DNA transposon families, Chapaev-2 and CEMUDR1 (Fig 3A and Appendix Fig S2A).

Since DNA transposons are the most abundant transposable elements in *C. elegans*, and thought to be the only class active in this organism (Bessereau, 2006; Laricchia *et al*, 2017), we hypothesized they might be targets of piRNA-mediated silencing. When we scanned these two transposons families for potential piRNA target sites (Bagijn *et al*, 2012), we found several matches. Chapaev-2 is a potential target of two piRNAs with up to two mismatches and ten piRNAs with up to three mismatches. CEMUDR1 is a potential target of one piRNA with perfect matching, five piRNAs with up to one mismatch, and 17 piRNAs with up to three mismatches (Fig 3B). Moreover, we also checked published data (Bagijn *et al*, 2012) for the density of endogenous 22G siRNAs antisense to these transposons in wild-type and *prg-1* animals and found that, for both transposons, 22G siRNAs were present at high levels in wild-type animals but were decreased in *prg-1* mutants. Together, these observations confirm that both Chapaev-2 and CEMUDR1 are piRNA pathway targets and indicate that *rpb-9* is required to repress their activity in the germline.

In *C. elegans*, the piRNA pathway has been previously implicated in the regulation of endogenous genes (Rojas-Ríos & Simonelig, 2018). For this reason, we also analyzed the expression of endogenous mRNA transcripts in *rpb-9* mutants. When we compared the polyA-selected transcriptome of *rpb-9* and wild-type animals, we found a total of 1,556 misregulated transcripts in *rpb-9* mutants, with 292 being downregulated (18.76%) and 1,264 upregulated (81.23%) (Fig 3C).

Since the piRNA pathway acts in the germline, we wondered whether some of the upregulated transcripts in *rpb-9* mutants corresponded to desilenced somatic genes in the germline. To test this, we took advantage of published data (Reinke *et al*, 2004; Bezler *et al*, 2019) to classify transcripts into "germline-specific" and "soma-specific" and used these classes to filter deregulated genes in *rpb-9* mutants. We found that, while germline-specific transcripts did not vary significantly, soma-specific transcripts tended to be upregulated in *rpb-9* mutants (Fig 3D). This suggests that, other than in controlling transposon activity, *rpb-9* might be involved in the repression of a subclass of somatic genes in the germline. Importantly, analysis of total RNA libraries yielded comparable results, both for transposable elements and for mRNA transcripts (Appendix Fig S2A–C).

In order to explore the relationship between *rpb-9* and the other major components of the piRNA pathway, we compared the transcriptomes of our *rpb-9* mutants with those of *prg-1* and *hrde-1* mutants. We observed that *rpb-9* and *prg-1* mutants share a total of 51 deregulated genes (22 upregulated and 29 downregulated, linear regression correlation coefficient = 0.03, hypergeometric test *P* value < 2.7e-11, representation factor = 2.8) (Fig 3E), while *rpb-9* and *hrde-1* share 42 (36 upregulated and 6 downregulated, linear regression correlation coefficient = 0.1, hypergeometric test *P* value < 1.9e-06, representation factor = 2.2) (Fig 3F). Importantly, we found the piRNA sensor transcript as upregulated in both *rpb-9* versus *prg-1* and *rpb-9* versus *hrde-1* comparisons, consistent with the idea that *rpb-9* represses germline transgenes via both the piRNA pathway and its downstream effector, the nuclear RNAi pathway. These observations suggest that *rpb-9* is required, together with *prg-1* and *hrde-1*, for gene silencing at a subset of piRNA pathway targets.

In order to refine this analysis, we decided to focus specifically on germline transcriptomes. By taking advantage of published datasets from gonad-dissected samples (Reed *et al*, 2020) and comparing them with our own, we asked what proportion of misregulated genes in *rpb-9* mutants could originate from within the germline. We observed a significant overlap between upregulated genes in *rpb-9* animals and upregulated genes in *prg-1* mutant germlines (525 genes, 40.8%, regression coefficient = 0.2161) (Fig 3G), in agreement with the hypothesis that transcriptome misregulations in *rpb-9* mutants mostly arise from germline-related piRNA pathway defects.

## Many upregulated genes show invariant or reduced RNA Pol II enrichment

We reasoned that the transcriptionally downregulated genes we identified above might be a result of a defect in the canonical role for *rpb-9* in promoting transcription as part of the RNA Pol II complex, consistent with ubiquitous *rpb-9* expression. Thus, in order to decipher the role of *rpb-9* in the context of the piRNA pathway, we decided to focus our attention on upregulated genes.

Given the reported role for *rpb-9* in TSS selection (Hull *et al*, 1995; Sun *et al*, 1996; Ghazy *et al*, 2004), we sought to understand whether upregulation/desilencing of these genes would merely depend on increased binding of RNA Pol II at their TSS, or whether another mechanism could be implicated. In order to explore these possibilities, we performed an RPB-1/AMA-1 chromatin immunoprecipitation followed by sequencing (ChIP-seq) analysis. We did not observe major differences in RNA Pol II binding genome wide (Fig EV2A), but noticed differential binding patterns for upregulated genes in *rpb-9* mutants compared with wild type. A certain proportion (138 genes, 33.3%) of the 384 upregulated genes with detectable RNA Pol II signal also showed increased (Mann–Whitney *U*-test, *P* = 2.06e-07) RNA Pol II binding at the TSS and within the gene body, compared to wild type (class I genes) (Fig 4A). Surprisingly, however, we observed that the majority of upregulated genes (246 genes, 66.7%) displayed unchanged (Mann–Whitney *U*-test, *P* = 0.77) or reduced (Mann–Whitney *U*-test, *P* = 0.0136) RNA Pol II binding in *rpb9* animals, despite being upregulated (or desilenced) (class II (Fig 4B) and class III genes (Fig 4C)). Importantly, the piRNA sensor belonged to class III genes (Fig 4D). Analysis of total RNA libraries yielded similar results (Fig EV2B–D).

We also examined Pol II binding to the upregulated Chapaev-2 and CEMUDR-1 transposons, and observed that it was decreased in *rpb-9* mutants compared with wild type, similarly to the piRNA sensor (Fig 4E).

These results suggested an additional role for *rpb-9*, possibly uncoupled from its canonical function as RNA Pol II subunit at a subset of endogenous genes. As control, we additionally analyzed Pol II profiles over downregulated genes and observed that the majority of them (94%) displayed reduced (Mann–Whitney *U*-test, *P* = 7.26e-5) Pol II binding in *rpb-9* mutants, as expected given the canonical role of *rpb-9* in transcription (Fig EV2E).

In order to elucidate novel *rpb-9* nuclear functions, we set out to identify its protein interactors. To be able to perform an immunoprecipitation followed by mass spectrometry (IP-MS), we generated an endogenously tagged *rpb-9::ollas* strain using the CRISPR/Cas9 system (Jiang & Marraffini, 2015; Akay *et al*, 2017). RPB-9::OLLAS is expressed (Fig EV3A) and does not induce piRNA sensor desilencing (Fig EV3B), suggesting that the protein fusion is functional.

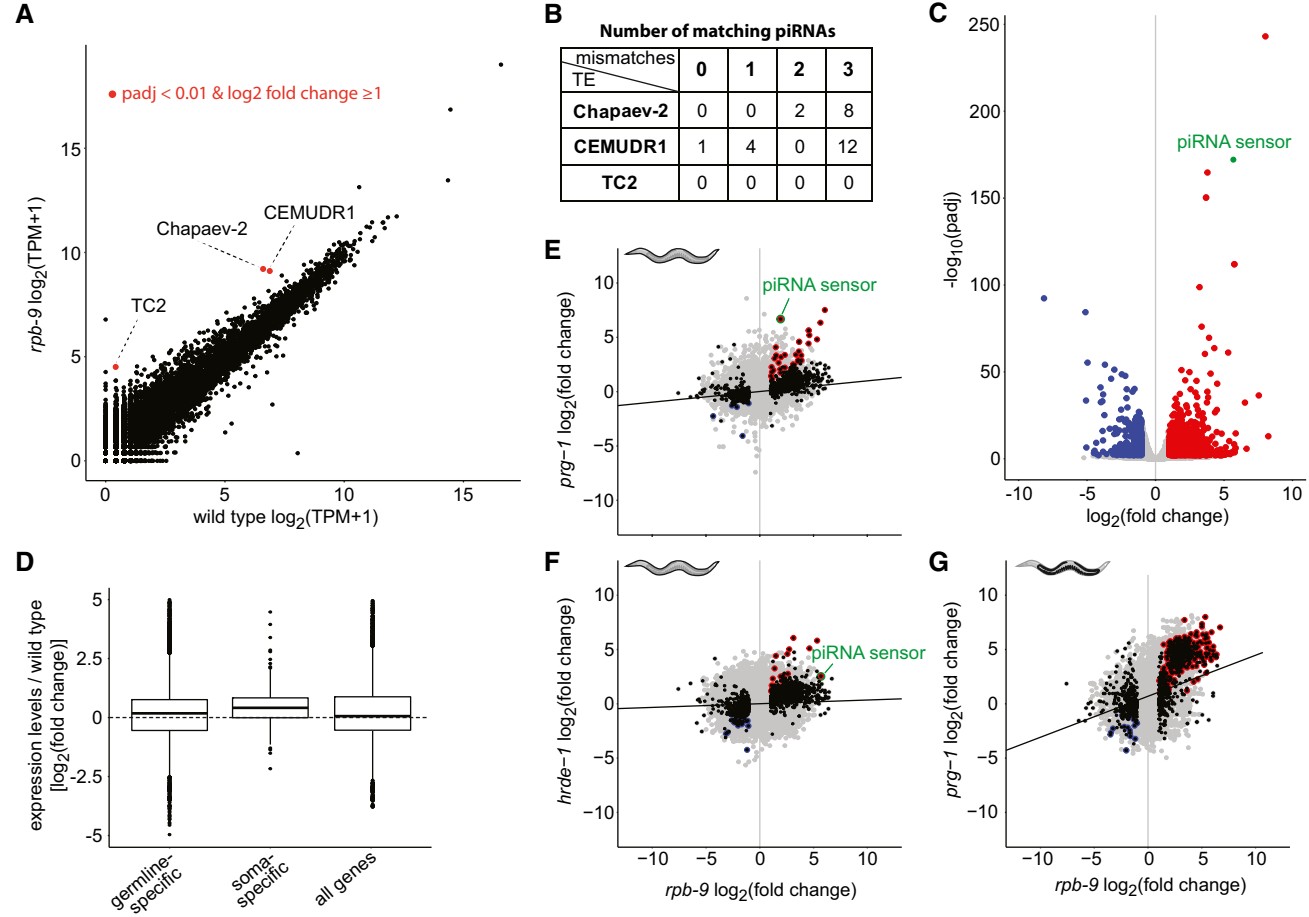

**Figure 3. *rpb-9* represses two DNA transposon families and a subset of somatic genes.**

A    Differential expression analysis of transposable elements in *rpb-9* (*mj261*) mutants versus wild type (polyA-selected RNA-seq libraries) [$P \leq 0.01$, log₂(fold change) $\geq 1$, DESeq2 analysis]

B    Table showing the numbers of matching piRNAs (with 0, 1, 2, and 3 mismatches, respectively) for each of the upregulated transposon families.

C    Genome-wide differential gene expression analysis of *rpb-9* (*mj261*) mutants versus wild type (polyA-selected RNA-seq libraries) [$P \leq 0.01$, log₂(fold change) > 1 or < −1, DESeq2 analysis].

D    Differential expression analysis of germline-specific and soma-specific genes, as classified in Reinke *et al* (2004). "All genes" is shown for comparison (polyA-selected RNA-seq libraries, $n = 3$).

E–G  Pairwise correlation plots of *rpb-9* (*mj261*) and *prg-1* (*n4357*) (E), *hrde-1* (*tm1200*) (F), or gonad-dissected *prg-1* (*n4357*) (Reed *et al*, 2020) (G) transcriptomes. Differentially expressed genes in *rpb-9* (*mj261*) are shown in black, shared upregulated genes in red, and shared downregulated genes in blue. The piRNA sensor transcript is highlighted in green (total RNA "Ribo-Zero" RNA-seq libraries).

In the pull-down experiment, we found a total of 25 proteins with significantly enriched binding compared to the control (Fig 5A, Welch's *t*-test, $P < 0.05$). Four of them were other polymerase subunits: RPB-1/AMA-1, already reported to be associating with RPB-9 to form the DNA-binding groove of the holoenzyme in human (Acker *et al*, 1997), and RPB-2, RPB-5, and RPB-7, all contacts that were never reported before. Interestingly, other proteins were factors involved in transcription elongation and termination, including the elongation factors SPT-5 (a component of the DSIF complex), CTR-9 (a component of the PAF-1 complex), and EMB-5 (human SUPT6h homolog), and the early-termination factor NRD-1 (human SCAF4 homolog). The identity of these binding partners is consistent with a previously described role for Rpb9 in transcriptional elongation in

yeast (Awrey *et al*, 1997; Hemming *et al*, 2000; Van Mullem *et al*, 2002).

Given these results, we hypothesized that *rpb-9* could be important for transcriptional elongation of class II and class III genes, and we speculated that premature termination of transcriptional elongation could manifest in *rpb-9* mutants and result in shorter nascent transcripts and hence in a limited docking platform for the HRDE-1/22G-siRNA silencing machinery. In order to test this hypothesis, we designed primers spanning three intron–exon junctions along the *gfp* transgene within the piRNA sensor and measured unspliced *gfp* levels in *rpb-9* mutants and wild-type animals by real-time qPCR. The results show that, although the unspliced transcript is significantly upregulated in *rpb-9* mutants, the ratios of unspliced *gfp* levels along the transcript (3′ end versus 5′ end) are not significantly

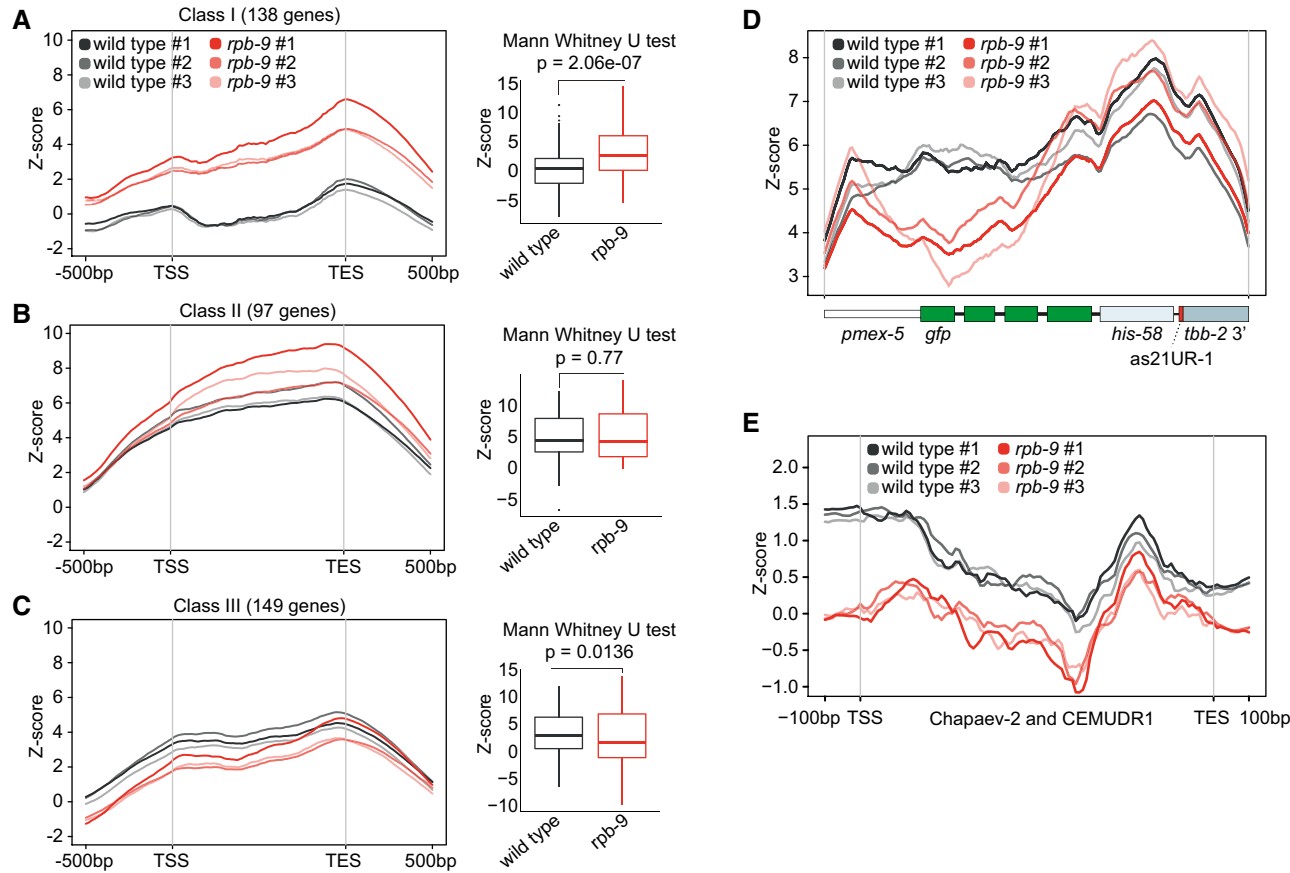

**Figure 4. *rpb-9* affects RNA pol II binding to a subset of genes.**

A–C Analysis of RNA Pol II binding (RPB-1/AMA-1 ChIP-seq) at upregulated genes in *rpb-9 (mj261)* mutants, as defined in Fig 3C. Class I: upregulated genes with increased RNA Pol II binding (A); class II: upregulated genes with invariant RNA Pol II binding (B); class III: upregulated genes with reduced RNA Pol II binding (C). n = 3 biological replicates are shown per each genotype (polyA-selected RNA-seq libraries). Central bands represent the median, boxes represent the 25th and 75th percentiles, and whiskers represent the lowest and highest values, excluding outliers.

D RNA Pol II binding profile over the piRNA sensor locus.

E RNA Pol II binding profile over the Chapaev-2 and CEMUDR1 DNA transposons, scaled to 1kb.

different compared with wild type, suggesting that transcriptional elongation at the piRNA sensor locus is in fact efficient in *rpb-9* animals (Fig 5B). Taken together, these results suggest that the establishment of piRNA-dependent silencing in *rpb-9* mutants is defective upstream of nascent transcript synthesis.

**rpb-9 is required for robust secondary piRNA pathway-dependent HRDE-1-associated 22G siRNA production**

Next, we tested whether the defect in piRNA target silencing in *rpb-9* mutants was caused by a decrease in the abundance of endogenous secondary 22G siRNAs. Indeed, a strong reduction in the amount of these molecules could explain why reduced or unchanged RNA Pol II binding on class II and class III genes (with wild-type lengths of nascent transcripts) still results in their upregulation/desilencing.

Endogenous secondary 22G siRNAs are not only mediators of silencing, but they can also counteract it. This is for example the case for CSR-1-associated 22G siRNAs, which are the effectors of

a protection mechanism against the piRNA-mediated silencing of germline-expressed genes (Wedeles *et al*, 2013; Seth *et al*, 2013). We reasoned that, for these targets, high 22G siRNA amounts would correlate with higher gene expression rates. On the other hand, piRNA target genes would display high 22G siRNA amounts but low expression rates. For these reasons, we explored the correlation between mRNA expression levels and endogenous secondary 22G siRNA abundance genome wide. We subdivided the transcriptome into 20 bins (A to T), each containing 857 genes, which were ranked according to their 22G siRNA density in wild type (Fig 5C). We observed a general trend where 22G siRNA density correlated with mRNA expression levels, consistent with the idea that the more an mRNA is expressed, the more 22G siRNAs can be produced for that locus. A notable exception was the last bin (T), corresponding to the genes with the highest 22G siRNAs density (top 5%), but which displayed lower mRNA expression levels. This pattern is consistent with active 22G siRNA-mediated silencing at high 22G siRNA densities. Interestingly, the mean expression levels of genes in bin T were higher

in *rpb-9* mutants compared with wild type. This suggests that these genes are direct piRNA targets. In further support of this hypothesis, we found that the piRNA sensor belongs to this subset of genes (bin T). Finally, we observed that the highest proportion of piRNA targets (with ≤ 1 mismatch (Bagijn *et al*, 2012)) belong to this bin (Fig 5C and D).

We next explored the correlation between transposable element expression and endogenous 22G siRNA abundance. We generated 10 bins (A–J), each containing 764 elements, and observed that the upregulated transposons Chapaev-2 and CEMUDR1 were included in the last bin (J), which contains transposons with high 22G density, but lower expression (top 10%) (Fig 5E). This suggests that

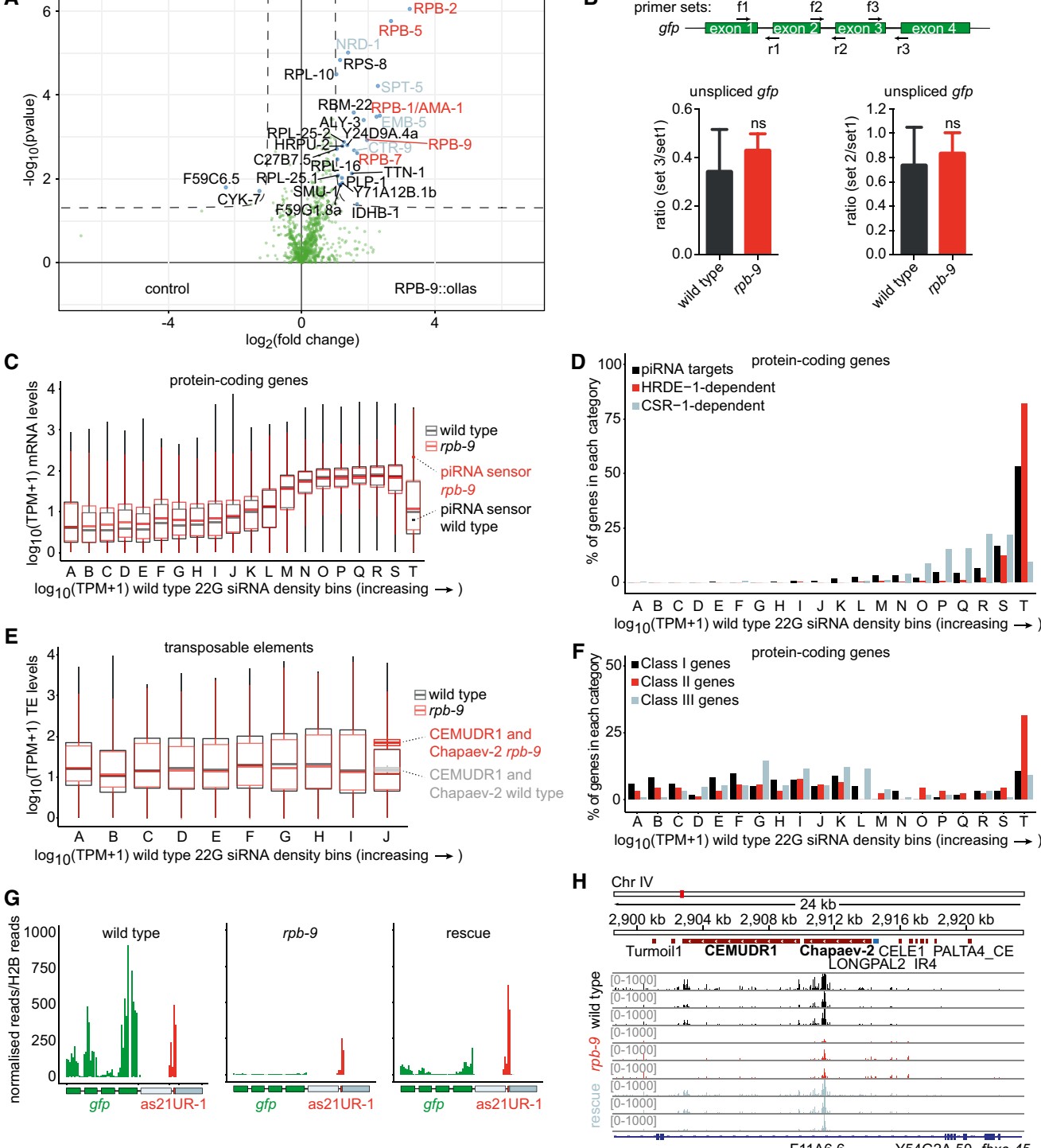

**Figure 5.**

**Figure 5.** *rpb-9* is required for the robust production of piRNA pathway-dependent HRDE-1-associated secondary 22G siRNAs.

A  Enrichment analysis of RPB-9-bound proteins in *rpb-9::ollas (mj604)* versus wild type by IP/MS [$P \le 0.05$, $\log_2$(fold change) > 1 or < −1, Welch's *t*-test]. $n = 4$ technical replicates (pooled).
B  Transgene transcript length is not affected in *rpb-9 (mj261)* mutants. Scheme of the *gfp* unspliced transcript showing the positions of RT–qPCR primers (top) and pairwise ratios (3′ to 5′) of unspliced *gfp* amounts along the transcript as indicated by the primer pairs (bottom) ($n = 3$, error bars represent SD). ns = not significant (two-tailed *t*-test).
C  Transcriptome binning according to increasing 22G siRNA density in wild-type animals (gray, $n = 3$). Mean normalized *rpb-9 (mj261)* mRNA reads (red, $n = 3$) are overlaid with mean normalized wild-type mRNA reads (gray) (polyA-selected RNA-seq libraries). The piRNA sensor transcript is indicated with red (*rpb-9 (mj261)*) and gray (wild-type) dots. Central bands represent the median, boxes represent the 25th and 75th percentiles, and whiskers represent the lowest and highest values, excluding outliers.
D  Distribution of piRNA targets (as defined in Bagijn *et al*, 2012) and of HDRE-1- and CSR-1-dependent 22G siRNAs across bins as defined in (C).
E  TE binning according to increasing 22G siRNA density in wild-type animals (gray, $n = 3$). Mean normalized *rpb-9 (mj261)* mRNA reads (red, $n = 3$) are overlaid with mean normalized wild-type mRNA reads (gray) (polyA-selected RNA-seq libraries). The CEMUDR1 and Chapaev-2 transcripts are indicated with red (*rpb-9 (mj261)*) and gray (wild-type) box plots. Central bands represent the median, boxes represent the 25th and 75th percentiles, and whiskers represent the lowest and highest values, excluding outliers.
F  Distribution of class I, class II, and class II genes (as defined in Fig 4A–C) across bins as defined in (C).
G  Distribution of normalized 22G siRNA reads mapping over the piRNA sensor.
H  Distribution of normalized reads mapping over the CEMUDR1 and Chapaev-2 DNA transposons.

Chapaev-2 and CEMUDR1, similarly to the piRNA sensor, are direct piRNA targets.

Since different AGO proteins bind to different subsets of 22G siRNAs and mediate different outcomes with regard to gene silencing and gene licensing, we then examined how HRDE-1- and CSR-1-dependent 22G siRNAs were distributed across the mRNA bins (Fig 5D). We observed that, where 22G siRNA abundance and mRNA expression correlated the most (bins N to S), CSR-1-associated 22G siRNAs were present at the highest proportions and the mean expression of the corresponding genes did not change significantly in *rpb-9* mutants (two-sample *t*-test, $P < 0.05$). Conversely, where 22G siRNA abundance and mRNA expression were anticorrelated (high 22G siRNAs but low mRNA expression) (bin T), HRDE-1-associated 22G siRNAs were found in the highest proportions and the mean expression of the corresponding genes was higher in *rpb-9* mutants compared with wild type. These observations support the hypothesis that the genes belonging to this class are direct targets of piRNAs dependent on *rpb-9* function. Interestingly, when we analyzed how class I, class II, and class III genes were partitioned across the bins, we observed that class II (and to a minor extent class III) genes mostly resided in the last bin (T), in agreement with our hypothesis of them being direct downstream targets of *rpb-9*-dependent piRNAs (Fig 5F). Importantly, these observations held true also for the analysis of total RNA libraries (Fig EV3C–E).

As the piRNA sensor belonged to bin T, we examined the distribution of antisense 22G siRNAs mapping to it more closely. We observed that, while *gfp* 22G siRNAs mapping to the piRNA sensor were present at high levels over all exons in wild-type animals, they were mostly depleted in *rpb-9* mutants and partially recovered in the rescue strain (Fig 5G), indicating that *rpb-9* is indeed required to induce the generation of sufficient secondary 22G siRNAs to repress the sensor. We observed a similar trend in secondary 22G siRNAs mapping across the two desilenced transposons Chapaev-2 and CEMUDR1, confirming that *rpb-9* is required to suppress their expression via the piRNA-dependent generation of 22G siRNAs (Fig 5H).

Next, we quantified the density of 22G siRNAs mapping over all piRNA targets (Bagijn *et al*, 2012) and observed that many of these transcripts (33%, 64 genes) were depleted in antisense 22G siRNAs in *rpb-9* mutants compared with wild type (hypergeometric test,

$P < 0.017$) (Fig EV3F). This confirms that *rpb-9* is required for the production of 22G siRNAs at a subset of piRNA targets. We observed a similar phenotype in *hrde-1* mutants, although the affected targets did not overlap completely with those of *rpb-9* animals, in agreement with our transcriptome correlation analysis (Fig 3F). Consistent with a prominent role for *prg-1* in the generation of piRNAs, *prg-1* mutants showed an even stronger depletion in 22G siRNAs at piRNA targets. Importantly, the defect of *rpb-9* mutants was reverted in the *rpb-9* rescue strain (Fig EV3F). Together, these data indicate that *rpb-9* is required to produce levels of HRDE-1-bound 22G siRNAs that are sufficient to efficiently establish silencing at piRNA target loci.

### *rpb-9* is required for the generation of mature piRNAs

In wild-type animals, high levels of endogenous 22G siRNAs are part of a self-sustaining response initiated by RdRPs upon binding of primary siRNAs, including piRNAs, to complementary transcripts. We therefore decided to measure the amounts of mature piRNAs in *rpb-9* mutants.

First, we analyzed global mature piRNA levels and found a significant reduction in their expression in *rpb-9* mutants, which was restored in the rescue strain (Fig 6A, two-sample *t*-test $P < 0.05$). Although this reduction was not as strong as in *prg-1* mutants (Suen *et al*, 2019), it suggests that *rpb-9* is required to produce mature piRNAs. We next wondered whether this reduction in expression levels affected all piRNAs or only a subset of them. We thus plotted piRNA expression data along chromosome coordinates in equal bins and observed that all the piRNA species encoded in the two chromosome IV clusters were downregulated in *rpb-9* mutants compared with wild type (Fig 6B) but were restored in the rescue (Fig EV4A). This suggests that *rpb-9* controls piRNA production via a single mechanism, possibly independent of sequence variations among single piRNAs.

We also quantified the abundance of piRNA 21UR-1, whose complementary sequence is present within the piRNA sensor. We observed a subtle reduction in the mature levels of this piRNA in *rpb-9* mutants compared with wild type, both by Northern blotting (Fig 6C) and by RT–qPCR (Fig EV4B). This is likely the primary cause of the observed loss of 22G siRNAs. In agreement

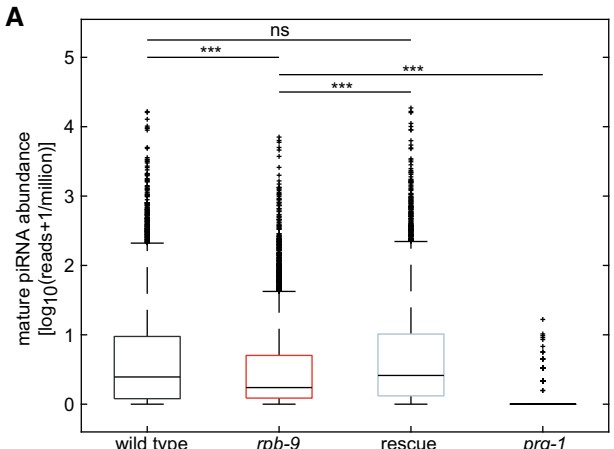

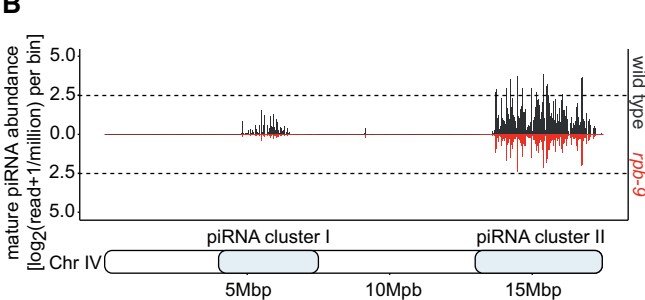

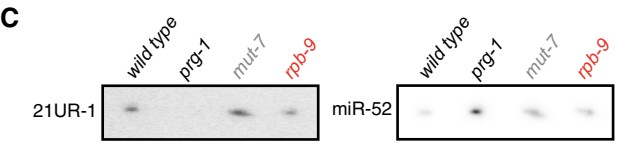

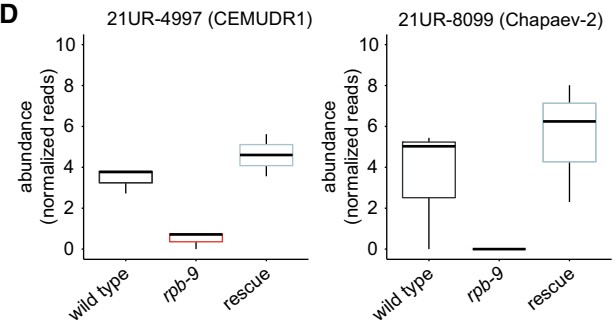

**Figure 6. rpb-9 is required for the generation of mature piRNAs.**

A Expression levels of mature piRNAs in wild-type, *rpb-9 (mj261)*, *rpb-9* rescue *(mj261; mjSi70)*, and *prg-1 (n4357)* animals (*n* = 3, hypergeometric test). Central bands represent the median, boxes represent the 25th and 75th percentiles, and whiskers represent the lowest and highest values, excluding outliers.

B Mature piRNA expression along chromosome IV coordinates (motif-dependent piRNA clusters I and II) in wild-type and *rpb-9 (mj261)* animals.

C Northern blot quantification of mature piRNA 21UR-1 levels in wild-type, *prg-1 (n4357)*, *mut-7 (mj253)*, and *rpb-9 (mj261)* animals.

D Quantification of the levels of the mature piRNAs 21UR-4997 and 21UR-8099 by small RNA sequencing (raw reads normalized to library size, *n* = 3). Central bands represent the median, boxes represent the 25th and 75th percentiles, and whiskers represent the lowest and highest values, excluding outliers.

with this hypothesis, we observed that, although some 22G siRNAs were still detectable over the 21UR-1 target site within the piRNA sensor (Fig 5G), their levels were lower than wild type, and 22G siRNA spreading to the rest of the transcript was defective. We observed this previously in other piRNA pathway mutants (Akay *et al*, 2017).

Similarly, the amounts of some of the piRNAs predicted to target the Chapaev-2 and CEMUDR-1 transposons were decreased in *rpb-9* mutants (Fig 6D), confirming that *rpb-9* represses them via the piRNA pathway. All together, these data indicate that *rpb-9* is required to generate wild-type levels of mature piRNAs.

## RPB-9 recruits Integrator at motif-dependent piRNA loci to terminate transcription

We reasoned that this decrease in the abundance of mature piRNAs in *rpb-9* mutants would originate from a defect in the production of the corresponding piRNA precursors, so we decided to investigate the role of *rpb-9* in piRNA loci transcription.

In order to capture all piRNA precursors, we sequenced short capped RNAs from isolated germ nuclei obtained from young adult animals, after separation of chromatin-bound (nascent) and nucleoplasmic fractions (as described in Beltran *et al*, 2020).

We found that the amount of both chromatin-bound (Fig EV5A) and nucleoplasmic (Fig EV5B) motif-dependent piRNA precursors tended to be lower in *rpb-9* mutants compared with wild type, perhaps explaining the observed decrease in the levels of motif-dependent mature piRNAs (Fig 6B) and suggesting that *rpb-9* could be required for motif-dependent piRNA loci transcription. We did not observe this trend for motif-independent piRNA precursors, indicating that *rpb-9* does not appear to be required for their transcription (Fig EV5C and D).

The process of transcription depends on the tight regulation of initiation, elongation, and termination. Previous studies in yeast have shown that Rpb9 both modulates the selection of the TSS and is involved in the elongation of transcription: In cells lacking Rpb9, the population of TSSs is shifted upstream at a subset of promoters (Furter-Graves *et al*, 1994; Hull *et al*, 1995; Sun *et al*, 1996), and in *in vitro* elongation assays, an RNA Pol II complex lacking Rpb9 pauses at intrinsic elongation blocks at a lower frequency compared with wild type (Hemming *et al*, 2000).

In order to understand whether *rpb-9* was required for precise initiation and/or elongation of transcription at piRNA genes, we examined the length of all piRNA precursors. We noticed that the median precursor length in *rpb-9* mutants was higher than the median in wild type for both nascent and nucleoplasmic species and that this defect was reverted in the rescue (Figure 7A). In wild-type animals, the length distribution of nascent piRNA precursors is bimodal, with peaks at 26 nt and 45 nt (Fig 7B and C, Beltran *et al*, 2020). In *rpb-9* mutants, the length distribution is still bimodal, but both peaks are shifted to ~28 nt and ~47 nt, respectively (Fig 7B and C). Similarly, also the length distribution of nucleoplasmic piRNA precursors is shifted in *rpb-9* mutants (Fig 7D). In both cases, the length defect is recovered in rescue animals (Fig 7B–D).

Drawing a parallel from the aforementioned observations in yeast, the presence of longer piRNA precursors in *rpb-9* mutants could be a consequence of the selection of an upstream TSS or of a

defect in elongation/termination. In order to test whether TSS selection at piRNA loci depends on RPB-9, we analyzed the distribution of the 5′ ends of short capped RNA reads. For both wild-type and *rpb-9* animals, as well as for rescue individuals, we observed an enrichment of reads initiating 2 nt upstream of piRNAs U sites, suggesting that TSS selection is not impaired in *rpb-9* mutants and

that the increase in piRNA precursor length is rather due to a defect in 3′ end formation (Fig 7E).

At piRNA loci, termination of transcription is associated with promoter-proximal RNA Pol II pausing at AT-rich "termination signals" (Beltran *et al*, 2019) and depends on the 3′ end cleavage activity of the Integrator complex (Beltran *et al*, 2020). In order to

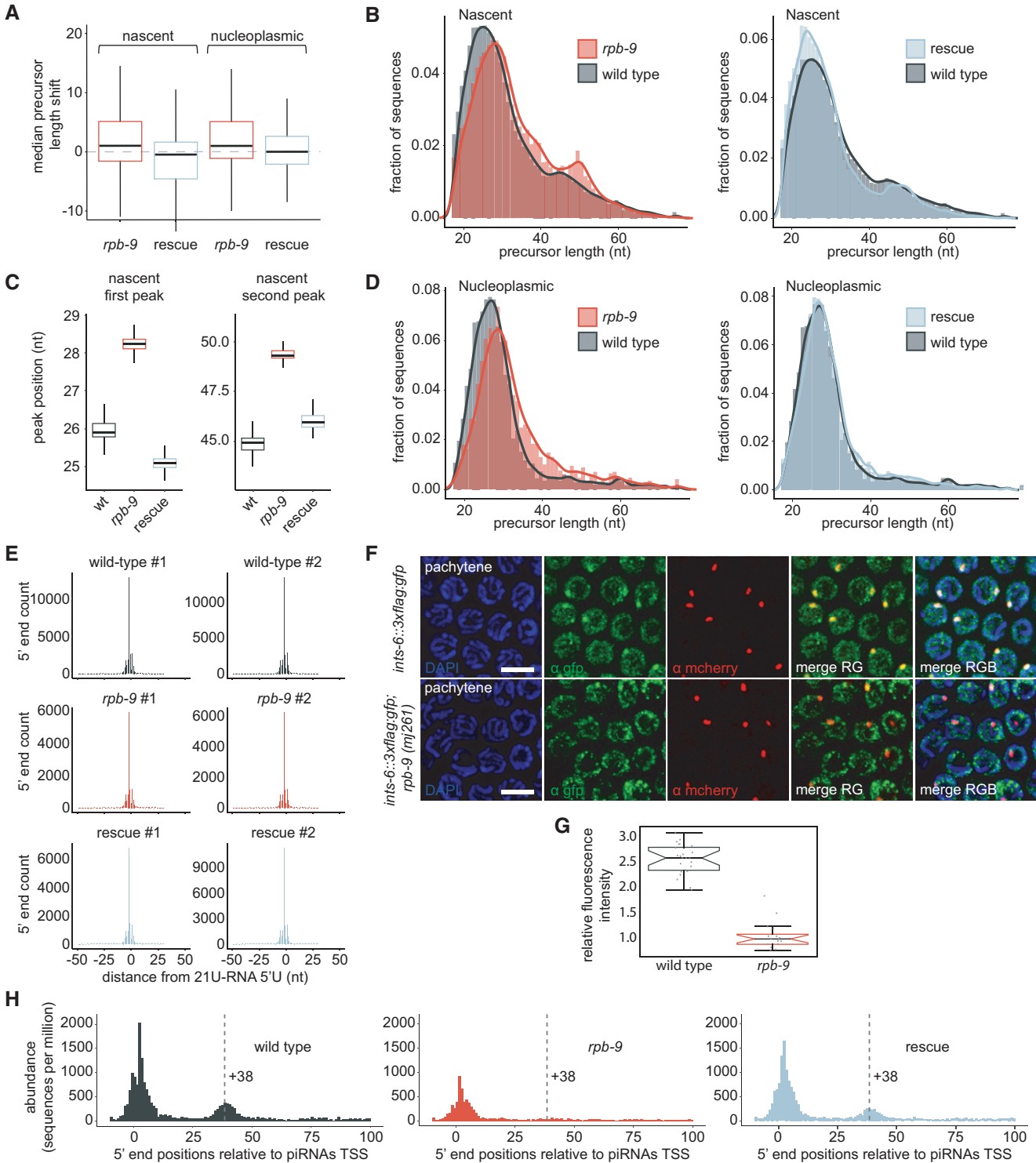

**Figure 7.**

**Figure 7. *rpb-9* is required for transcription termination at piRNA loci.**

A   Median precursor length shift of nascent (left) and nucleoplasmic (right) piRNA precursors in *rpb-9 (mj261)* mutants and rescue *(mj261; mjSi70)* animals compared with wild type (*n* = 2). Central bands represent the median, boxes represent the 25th and 75th percentiles, and whiskers represent the lowest and highest values, excluding outliers.

B   Distribution of nascent piRNA precursor lengths in *rpb-9 (mj261)* and rescue *(mj261; mjSi70)* animals compared with wild type.

C   Peak positions of nascent piRNA precursor length distributions in 2,000 subsamples of 5,000 precursor sequences sampled without replacement. Central bands represent the median, boxes represent the 25th and 75th percentiles, and whiskers represent the lowest and highest values, excluding outliers.

D   Distributions of nucleoplasmic piRNA precursor lengths in *rpb-9 (mj261)* and rescue *(mj261; mjSi70)* animals compared with wild type.

E   Distribution of 5′ ends of short capped RNA reads mapping to motif-dependent piRNA loci. Total counts of read 5′ ends mapping within a 50-bp window of motif-dependent piRNA loci, aggregated by position relative to 21U-RNA 5′ U sites. A comparable enrichment of reads initiating 2 nt upstream of 21U-RNA 5′ U sites is observed across replicates and genotypes.

F   *rpb-9* recruits Integrator at piRNA loci. Representative single-plan immunofluorescence images of Integrator (ints-6::GFP) and PRDE-1 (prde-1::mCherry) localization in wild-type and *rpb-9 (mj261)* animals. Scale bar = 5 μm.

G   Co-localization analysis of PRDE-1::mCHERRY and INTS-6::GFP foci in wild-type and *rpb-9* (mj261) animals. Central bands represent the median, notches represent the confidence interval around the median, boxes represent the 25th and 75th percentiles, and whiskers represent the lowest and highest values, excluding outliers. *n* (nuclei/genotype) > 16 (Welch's *t*-test, *P* = 1.7e-17).

H   Positions of unique 5′ monophosphate small RNA reads mapping at piRNA promoters after removal of reads > 15 nt initiating at annotated piRNA 5′ U sites. The average signal is normalized to sequences per million of mapped reads.

understand whether *rpb-9* could be required to recruit Integrator at piRNA genes to promote termination, we crossed our *rpb-9* mutants with animals in which the core Integrator subunit INTS-6 and the piRNA biogenesis factor PRDE-1 are endogenously tagged with GFP (Gómez-Orte *et al*, 2019) and mCherry (Weick *et al*, 2014), respectively, and analyzed Integrator localization via immunostaining. In wild-type animals, the Integrator complex localizes throughout the nucleus, but accumulates specifically on piRNA genes, as indicated by the presence of INTS-6 clouds and their co-localization with PRDE-1 signals (Fig 7F and G, Beltran *et al*, 2020). In *rpb-9* mutants, Integrator is still homogeneously distributed in the nucleus, but the accumulation clouds are lost, and INTS-6/PRDE-1 co-localization is significantly decreased (Fig 7F and G) (*P* = 1.7e-17, Welch's *t*-test). This indicates that RPB-9 is required to physically recruit the Integrator complex at piRNA loci, where its cleavage activity is necessary to terminate transcription of piRNA precursors.

To confirm this conclusion, we examined our samples for the presence of ~20 nt long 3′ nascent RNA cleavage fragments, a signature of Integrator activity, downstream of 21U-RNAs. Consistent with a role for *rpb-9* in recruiting Integrator, we observed a marked decrease in the abundance of these fragments at the termination region (around the +38 nucleotide position from the TSS) in *rpb-9* mutants compared with wild type (Fig 7H).

In conclusion, these data indicate that RPB-9 is required to promote transcription termination at motif-dependent piRNA genes by physically recruiting the Integrator complex at the 3′ ends of nascent RNAs upon RNA Pol II backtracking.

# Discussion

Initiation of piRNA pathway-mediated repression depends on the faithful transcription of piRNA loci. Here, we have characterized the contribution of the RNA Pol II subunit RPB-9 to this process. We have shown that *rpb-9* is required to promote efficient transcription termination at motif-dependent piRNA loci by physically recruiting the Integrator complex and eliciting Integrator-dependent cleavage of the 3′ end of nascent transcripts, and demonstrated that this activity is required to exert proper repression of two DNA transposon families, as well as a subset of somatic genes, in the germline.

Mechanistically, a functional RPB-9 protein is required to produce a sufficient amount of mature piRNAs to ensure the generation of high levels of HRDE-1-dependent secondary 22G siRNAs, which target complementary transcripts and thereby induce their silencing (Fig 8).

Interestingly, a relatively modest decrease in mature piRNA levels in *rpb-9* mutants is sufficient to provoke a drastic reduction in the amounts of endogenous secondary 22G siRNAs antisense to piRNA target loci. This is consistent with the fact that the piRNA pathway relies heavily on amplification mechanisms and small RNA thresholds to establish silencing. Because of this, it is possible that genes targeted by low-abundance piRNAs are more sensitive to small changes in their amounts, and hence more susceptible to desilencing. Indeed, this seems to be the case for the Chapaev-2 and CEMUDR1 DNA transposon families, which are targeted by lowly-abundant piRNAs and show a strong desilencing phenotype in our mutants. Chapaev-2 and CEMUDR1 are also desilenced in *prg-1* (Bagijn *et al*, 2012) and *hrde-1* (Akay *et al*, 2017) mutants, in agreement with our conclusion that *rpb-9* functions upstream of both these components in the piRNA pathway. However, this is not the case in *prde-1* mutants; *prde-1*, like *rpb-9*, is also required for the transcription of motif-dependent piRNA loci (Weick *et al*, 2014). We speculate that this discrepancy is due to the fact that PRDE-1 and RPB-9 perform different molecular functions in the context of piRNA loci transcription. While PRDE-1 defines the site of piRNA precursor generation, thereby influencing transcription initiation, RPB-9 is required for transcription termination. Impaired 3′ end formation may affect different piRNA loci to different extents. As mentioned above, piRNA-dependent transposon repression relies on specific small RNA thresholds: In this particular case, even minor differences in the abundance of certain mature piRNAs in *rpb-9* versus *prde-1* mutants could result in profound differences in their effects on repression of the same target.

An interesting observation is that the termination defects of *rpb-9* mutants are not detectable at deregulated genes other than at piRNA loci, suggesting that RPB-9, a conserved RNA Pol II subunit, may fulfill a very specific role in *C. elegans*. Perhaps RPB-9 is part of an ancient network, which, together with SNPC-4 (Kasper *et al*, 2014) and the USTC complex (Weng *et al*, 2019), has been recruited to direct transcription at piRNA loci. The

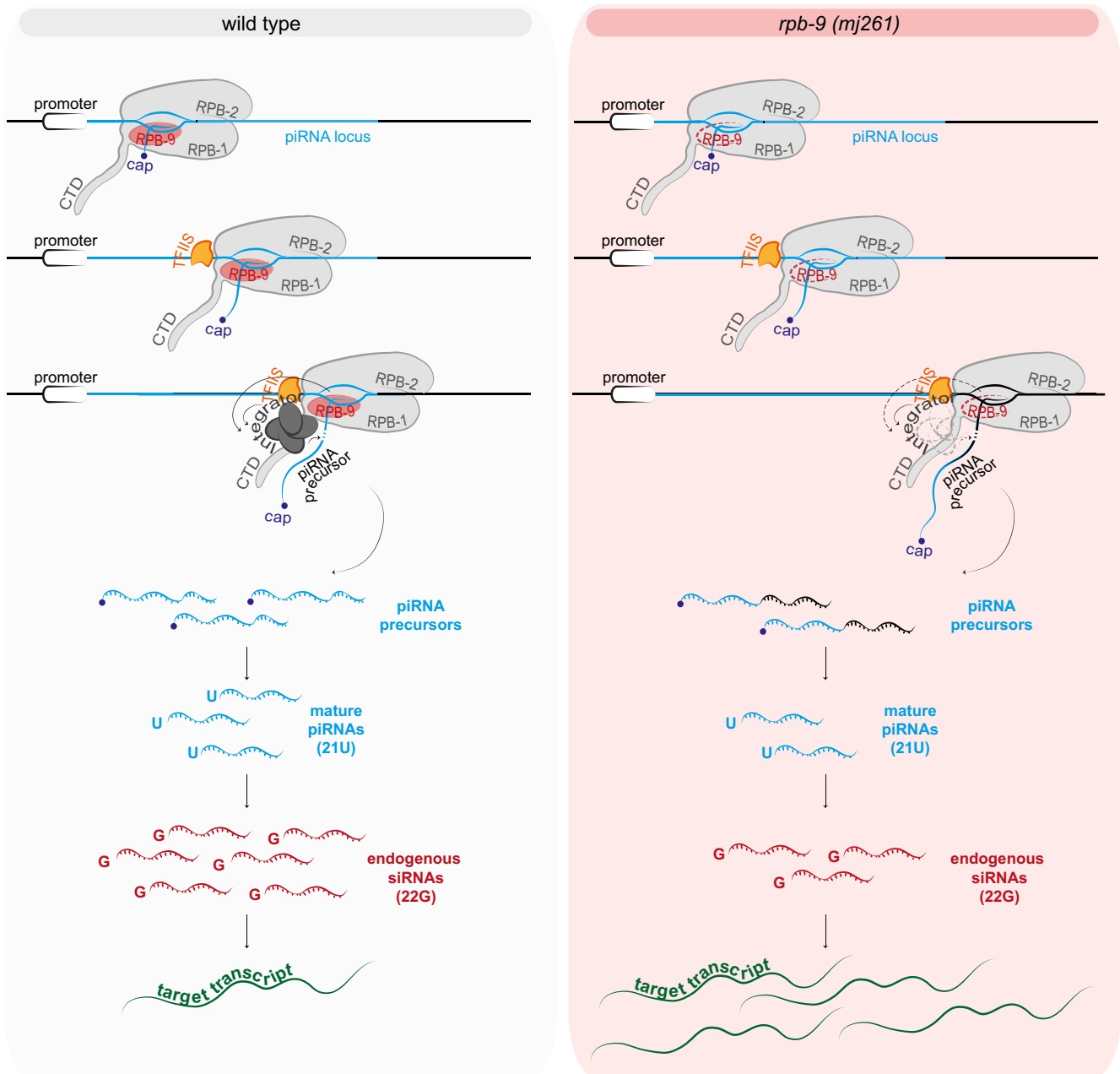

**Figure 8. Model of RPB-9 function.**

In wild-type animals, RPB-9 promotes Integrator-dependent cleavage of nascent motif-dependent piRNA precursors. piRNA precursors are then processed into mature piRNAs, which trigger a potent 22G siRNA amplification response. As a result, complementary target transcripts are silenced. In *rpb-9* mutants, transcription termination is defective and leads to longer piRNA precursors. As a consequence, the amount of mature piRNAs is decreased and the amplification response impaired. Target transcripts fail to be silenced.

recent discovery that motif-dependent piRNA units share evolutionary similarities with the highly conserved snRNA genes (Beltran *et al*, 2019), together with the fact that transcription of both these types of loci depends on the Integrator complex (Uguen & Murphy, 2003; Baillat *et al*, 2005; Ezzeddine *et al*, 2011; Ezzeddine *et al*, 2012) (Beltran *et al*, 2020), seem to favor this hypothesis. Whether *rpb-9* is required for transcription termination of snRNA genes in *C. elegans* is currently unknown, but we did not

observe a significant reduction in the expression levels of most snRNAs in *rpb-9* mutants, which indicates that RPB-9 is not essential for their transcription.

Studies in yeast have shown that Rpb9 is located at the tip of the "jaws" of the RNA Pol II complex, proposed to clamp the DNA downstream of the enzyme active site (Cramer *et al*, 2001; Gnatt *et al*, 2001). Despite this remarkable feature, deletion of Rpb9 in yeast does not lead to cell death (Woychik *et al*, 1991),

but only to minor transcriptomic changes in metabolism-related genes (Hemming *et al*, 2000), suggesting that Rpb9 is an accessory subunit in this organism. To understand whether this was also the case in *C. elegans*, we initially set out to generate complete knockout mutant animals via the CRISPR/Cas9 technology. Although we were able to observe heterozygous editing events upon sgRNA injections in the germline, our attempts at recovering *rpb-9* homozygous knockout mutants in the F2 progeny failed. This suggests that, similarly to rpb9 in *D. melanogaster*, *rpb-9* could in fact be essential in *C. elegans*. The mutant we discovered in our screen may produce either a truncated protein that retains some functionality or full-length protein at very low levels due to translational read-through at the predicted premature stop codon.

We have shown that *S. cerevisiae* Rpb9 and *C. elegans* RPB-9 share sequence identity at the two identified domains. In yeast, the N-terminal domain of Rpb9 has been shown to influence TSS selection, while the C-terminal domain has been reported to be involved in intrinsic transcript cleavage (Hemming *et al*, 2000). In our *rpb-9* mutants, a truncated but partially functional RPB-9 protein might explain why TSS selection is not affected, but transcriptional termination is. Indeed, the mutation we discovered in our screen is predicted to interrupt translation within the TFIIS C domain, but leaves the RPOL9 domain intact, suggesting that the both RPOL9 and TFIIS C domains may perform conserved functions in *S. cerevisiae* and *C. elegans*.

The conservation between the C-terminal domains of Rpb9 and tfiis in yeast has prompted *in vitro* studies with purified proteins. These early reports show that RNA Pol II ternary complexes lacking Rpb9 are defective in their response to TFIIS-stimulated read-through past an elongation block. They also suggest that the absence of Rpb9 does not affect the intrinsic cleavage ability of RNA Pol II nor its binding to TFIIS and that the addition of purified Rpb9 *in trans* restores the response of the complex to TFIIS (Awrey *et al*, 1997). It was thus proposed that Rpb9 might transmit a molecular signal from TFIIS to the RNA Pol II active site. Further genetic analyses in yeast supported this hypothesis (Hemming *et al*, 2000; Van Mullem *et al*, 2002). More recently, it has been shown that the association between Pol II and TFIIS is in fact perturbed in Rbp-9 mutant yeast (Sigurdsson *et al*, 2010).

We have shown here that the C-terminal domains of RPB-9 and TFIIS are conserved also in *C. elegans*. According to our study, RPB-9 is required to promote Integrator-mediated cleavage of the nascent transcript upon RNA Pol II backtracking at motif-dependent piRNA loci. Concomitantly, (Beltran *et al*, 2020) have reported a similar role for TFIIS. The fact that piRNA precursors are longer but can still be cleaved in *rpb-9* animals could perhaps indicate that TFIIS is still functional in these mutants, but not sufficient on its own to promote timely transcription termination. The same line of thought applies to *tfiis* mutants, which can terminate transcription, possibly thanks to a functional RPB-9 subunit, but do so in a defective manner (Beltran *et al*, 2020). Given these observations, we believe that it is possible that RPB-9 and TFIIS cooperate in the recruitment of Integrator activities to terminate transcription at piRNA loci. Supporting this hypothesis, offspring of double *tfiis;rpb-9* mutants display a maternal-effect embryonic-lethal phenotype (Appendix Fig S3). This result suggests that, indeed, *rpb-9* and *tfiis* are genetic interactors and

may play a role in the same pathway. The lethal phenotype, however, prevented further molecular analyses of this mutant.

Overall, our study sheds new light on how the piRNA pathway utilizes a core RNA Pol II subunit to guarantee high-fidelity transcription termination.

# Materials and Methods

### *C. elegans* culture

*Caenorhabditis elegans* were grown under standard conditions (Brenner, 1974) at 20 °C using the *Escherichia coli* strain HB101 as a food source. Genetic crosses were performed by mating males and females in a 5:1 ratio. All strains are listed in Appendix Table S1.

### *C. elegans* transgenics—CRISPR injections

RPB-9 C-terminus was tagged with an Ollas peptide by injecting wild-type N2 young adult animals with a mixture of target gene HR template (IDT oligos) (1 mg/ml), target gene CRISPR gRNA (Amersham Biosciences) (8 mg/ml), dpy-10 CRISPR gRNA (Amersham Biosciences) (2 mg/ml) as a marker, and His-Cas9 (in-house bacterial purification) (5 mg/ml) in a buffer comprising 10 mM KCL and 10 mM Tris–HCl pH: 8.0. Animals showing dumpy phenotype in the first generation after injection were screened for homozygous ollas tag insertion and backcrossed 3 times with the wild-type N2 strain to get rid of CRISPR off target effects. Genotyping primers and CRISPR oligos are listed in Appendix Table S2.

### *C. elegans* transgenics—mosSCI injections

For targeted single-copy transgene insertion, a mix of 20 ng/μl transgene construct, 50 ng/μl Mos transposase JL43, and 5 ng/μl for each co-injection marker (myo-2::mcherry::unc-54 and myo-3::mcherry::unc-54, or GFP markers as appropriate) was prepared and centrifuged for 30 min at 15,700 *g* at 4°C. Microinjection pads consisted of 2% agarose flattened onto a cover slip. Prior to microinjections, a pad was moistened by gently exhaling onto it, then a drop of halocarbon oil 700 (Sigma-Aldrich) was added to it. Young adult animals of the appropriate strain were transferred into the oil using an eyelash pick and flattened to the surface. Injections into gonad arms were performed using an Olympus IZ71 microscope equipped with an Eppendorf micromanipulator, FemtoJet injection rig, and Transfer Man joystick (Eppendorf). After injection, animals were left to recover in M9 medium, then transferred to individual plates, and left to recover overnight at 20°C. For selection of integrants, plates were placed at 25°C and offspring were assessed for moving unc-119 rescue animals after 3 and again after 4 days. Once plates with motile animals had starved, they were chunked onto a new plate and any moving animals lacking the co-injection marker were singled onto individual plates. A maximum of ten individual animals was picked from each parental plate. Offspring were screened or re-individualized for the absence of any Unc phenotype to generate homozygous lines. Transgene integration was validated by PCR of the insertion locus and transgene and, if applicable, by expression of a fluorescent protein. Details of the Mos system

and selection process are described in Frøkjaer-Jensen *et al* (2008, 2012).

### piRNA sensor EMS screen

After EMS treatment following standard protocols (Brenner, 1974), F2 or F3 offspring of mutagenized animals were sorted using a Copas Biosort large-particle sorter as described in Bagijn *et al* (2012). Further details are as described previously (Ashe *et al*, 2012). Chromosome mapping and genotyping of mutations are described in Weick *et al* (2014).

### Cloning of mosSCI plasmids

Plasmid constructs were generated performing Multi-Site Gateway Cloning (Invitrogen) according to manufacturer's instructions. pDONR entry constructs were made either by amplifying the gene of interest including exons from genomic DNA, or by amplifying the spliced transcript from wild-type cDNA. All pDONR entry clones were confirmed by sequencing. pDEST vectors after LR reactions were confirmed by colony PCR on corresponding bacterial clones and by expression of transgene after injection into *C. elegans*. pDEST *C. elegans* expression constructs are detailed in Frøkjær-Jensen *et al* (2012).

### piRNA sensor imaging

Representative single-plan images were acquired on a Leica SP8 confocal microscope at 63X magnification. Immobilized live animals were used.

### Transgenerational memory assay

Three L4 larvae per genotype were plated on gfp RNAi-expressing bacteria (5 replicates) or empty vector L4440 bacteria (3 replicates). G1 animals were analyzed under a fluorescence microscope, and one silenced animal per replicate per genotype was plated onto standard HB101 bacteria. At each generation, one silenced animal was singled from each plate to produce the next generation, and the remaining adult progeny was analyzed under a Kramer FBS10 fluorescence microscope. Animals were collected in M9, washed twice, quickly fixed in 70% ethanol, and deposited onto a glass slide coated with a 2% agarose pad. At least 50 animals per replicate per genotype were counted at each generation. Germline nuclear GFP brightness was categorically scored as on or off.

Representative images were taken on a DM6B fluorescence microscope (Leica) with a motorized stage and a Leica DFC9000 GT CCD camera. Exposures: DIC 25 msec, GFP 500 msec. 40× magnification.

### RNA extraction and real-time quantitative PCR of spliced and unspliced mRNAs

Total RNA was extracted using TRIzol reagent (Ambion, Life Technologies) and treated with Turbo DNase Kit (Invitrogen) according to the manufacturer's instructions. 500 ng of RNA was reverse-transcribed with random hexamers (Invitrogen) at 50°C for 1 h using SuperScript III reverse transcriptase (Thermo Fisher). Control reactions lacking enzymes were systematically run in parallel as negative controls. Real-time quantitative PCR was performed on 1 ml of diluted RT reactions (1/5) using SYBR Green Kit (Life Technologies) on a OneStepPlus Thermocycler (Thermo Fisher). All samples were run in duplicates. Expression levels were normalized to the reference gene *cdc-42*. Primer sequences are listed in Appendix Table S3.

### Cloning of the RPB-9 RNAi plasmid

The RPB-9 RNAi plasmid was constructed by cloning a 500-bp fragment containing most of the RPB-9 cDNA sequence into the L4440 RNAi feeding vector (Timmons & Fire, 1998) via Gibson assembly following the manufacturer's protocol. Briefly, 50 ng of the *rpb-9* insert was added to 50 ng of EcoRI-digested L4440 vector and incubated in Gibson master mix for 1 h at 50°C. 1 μl of the Gibson reaction was transformed into 50 μl of Dh5α chemically competent bacteria (NEB) and plated onto ampicillin-resistant plates. Resistant colonies were screened for correct insertions by the Sanger sequencing with the M13 forward primer and an in-house reverse primer (GGCCTTTTGCTCACATGTTC). The following sequence was synthesized as g-Block (IDT): **GAGACCGGCAGATCTGATATCATC GATG**TATCTGCAATTAAAATCAAAGCTTGAAAATGTTTTATCATA TTTTTTTCAGAAAGATGAGCCAAGGGTATGATAATTACGATGATAT GTACGATCAAAACGGTGCATCACCGGCGCCGAGTCAAAACGAAAA ACCCGGGAAAAGTGGGCCTGGTTTTGTTGGAATCAAGTTTTGCCC AGAATGCAATAATATGCTGTACCCACGAGAGGATAAGGAATCACG AGTTTTGATGTATTCCTGCCGGAACTGTGAGCATCGTGAAGTCGC CGCTAACCCGTGTATCTATGTGAATAAGCTCGTTCACGAAATTGA TGAGCTCACTCAAATCGTCGGAGATATTATTCACGATCCAACGCTC CCGAAGACTGAAGAACATCAATGTCCAGTCTGTGGCAAAAGTAAG GCTGTCTTCTTCCAGGCTCAAACAAAAAAGGCAGAAGAA**GAATTC GATATCAAGCTTATCGATACCGTC** (bold: L4440 homology arms, which flank an EcoRI restriction site in the backbone L4440 vector).

### Western blotting

Western blotting of RPB-9 in the *rpb-9 (mj261)* and *rpb-9* rescue *(mj261; mjSi70)* (Fig EV1A) was performed using a human anti-RPB-9 (NBP1-92344, Novus Biologicals) antibody and a mouse anti-tubulin (clone DM1A, Sigma) antibody.

Western blotting of RPB-9 in the *rpb-9::ollas* line (Fig EV3A) was performed using a mouse anti-tubulin (clone DM1A, Sigma) antibody and a rat anti-ollas (NBP1-06713, Novus Biologicals) antibody.

Protein samples were prepared by boiling approx. 50 adult animals per genotype in NuPAGE™ (Thermo Fisher Scientific) sample buffer according to the manufacturer's instructions. Denatured proteins were resolved on NuPAGE™ 4–12% Bis–Tris gradient gels (Thermo Fisher Scientific) and wet transferred on 0.45-μm pore-sized nitrocellulose membranes (Thermo Fisher Scientific) at 240 mAmp for 2 h. After overnight incubation with primary antibodies at 4°C, the membranes were incubated with HRP-conjugated secondary antibodies (GE Healthcare) for 1 h at room temperature. Chemiluminescence Reagents (Pierce ECL Thermo Scientific or Immobilon Western Millipore) were applied to the membranes, and the membranes were visualized on 18 × 24 cm X-ray films in a darkroom.

**RPB-9::ollas immunoprecipitation and mass spectrometry**

N2 control or RPB-9::ollas animals were grown to young adult stage, washed 3X with M9 buffer, and frozen in liquid nitrogen. Frozen *C. elegans* were crushed with a metallic grinder and lysed with a buffer containing 25 mM Tris–HCl pH: 7.5, 150 mM NaCl, 1.5 mM $MgCl_2$, 0.1% Triton X-100, and Complete Mini Protease inhibitor tablets (Roche, EDTA free). The lysate was sonicated using a Bioruptor with 10 cycles 30 s on/30 s off and then centrifuged for 30 min at 16,000 *g* (4°C) to remove insoluble material. BCA assay (Thermo Scientific) was used to determine total protein concentration of the supernatant. 4 mg total extract, 10 mg anti-ollas antibody (A01658-40, GeneScript), and 30 ml Protein A/G magnetic beads (Thermo Scientific) were used per technical replicate. In total, 4 technical replicates were used both for the N2 control and for RPB-9::ollas. After 12 h of incubation at 4°C in a rotating wheel, beads were washed 4 times with a buffer containing 25 mM Tris–HCl pH: 7.5, 150 mM NaCl, 1.5 mM $MgCl_2$, and Complete Mini Protease inhibitor tablets. Immunoprecipitates were then boiled with a 2X sample buffer for 15 min.

For the identification of RPB-9::ollas interactors, samples were separated on a 4%–12% NOVEX NuPAGE gradient SDS gel (Thermo) for 10 min at 180 V in 1× MES buffer (Thermo). Proteins were fixated and stained with Coomassie G250 Brilliant Blue (Carl Roth). The gel lanes were cut, minced into pieces, and transferred to an Eppendorf tube. Gel pieces were destained with a 50% ethanol/50 mM ammonium bicarbonate (ABC) solution. Proteins were reduced in 10 mM DTT (Sigma-Aldrich) for 1 h at 56°C and then alkylated with 5 mM iodoacetamide (Sigma-Aldrich) for 45 min at room temperature. Proteins were digested with trypsin (Sigma) overnight at 37°C. Peptides were extracted from the gel by two incubations with 30% ABC/acetonitrile and three subsequent incubations with pure acetonitrile. The acetonitrile was subsequently evaporated in a concentrator (Eppendorf) and loaded on StageTips (Rappsilber *et al*, 2007) for desalting and storage.

For mass spectrometric analysis, peptides were separated on a 20-cm self-packed column with 75 µm inner diameter filled with ReproSil-Pur 120 $C_{18}$-AQ (Dr.Maisch GmbH) mounted to an EASY HPLC 1000 (Thermo Fisher) and sprayed online into an Q Exactive Plus mass spectrometer (Thermo Fisher). We used a 94-min gradient from 2 to 40% acetonitrile in 0.1% formic acid at a flow of 225 nl/min. The mass spectrometer was operated with a top 10 MS/MS data-dependent acquisition scheme per MS full scan. Mass spectrometry raw data were searched using the Andromeda search engine (Cox *et al*, 2011) integrated into MaxQuant suite 1.5.2.8 (Cox & Mann, 2008) using the UniProt *C. elegans* database (August 2014; 27,814 entries). In both analyses, carbamidomethylation at cysteine was set as fixed modification, while methionine oxidation and protein N-acetylation were considered as variable modifications. Match-between-run option was activated. Prior to bioinformatic analysis, reverse hits, proteins only identified by site, protein groups based on one unique peptide, and known contaminants were removed.

For the further bioinformatic analysis, the LFQ values were log2-transformed and the median across the replicates was calculated. This enrichment was plotted against the – log 10-transformed *P* value (Welch's *t*-test) using the ggplot2 package in the R environment.

**AMA-1 ChIP-seq**

300,000 synchronized animals were grown to young adult stage on 140 mm NGM plates and collected in an M9 buffer per replicate (3 for wild type and 3 for rpb-9 mutant). Animals were washed in the same buffer three times. Animals were frozen in liquid nitrogen and then grinded with a metallic mortar. These extracts were fixed with 1% formaldehyde at room temperature 9 min. The crosslinking was quenched with the addition of 0.125 M glycine at RT for 5 min. for 4 min at 4,000 *g* (4°C). Following two washes, a final wash with a buffer containing 150 mM NaCl, 50 mM HEPES/KOH, 1 mM EDTA, 1% Triton X-100, 0.1% sodium deoxycholate, protease, and phosphatase inhibitors (Roche). Then, the samples were sonicated for 30 cycles 30 s on/30 s off. Afterward, the samples were centrifuged for 30 min at 15,700 *g* (4°C). 1 mg of the supernatant was used for ChIP with 4 µg of AMA-1 antibody (rabbit polyclonal, Novus, SDQ2357) incubated overnight rotating at 4°C together with 80 µl magnetic protein A/G beads (Thermo Scientific). Samples were then washed twice in 150 mM NaCl buffer, once with 500 mM NaCl buffer, once with 1 M NaCl buffer, once with TEL buffer (0.25 M LiCl, 1% NP-40, 1% sodium deoxycholate, 1 mM EDTA, 10 mM Tris–HCl pH 8, and freshly added protease inhibitor), and finally twice in TE buffer (pH 8). Beads were then eluted twice in 60 µl ChIP Elution Buffer (1% SDS, 250 mM NaCl in TE pH 8) at 65°C. Eluates of same samples and input were treated with 2 µl RNAse at 37°C for 1 h and then with 1 µl Proteinase K at 65°C to de-crosslink overnight.

**Library preparation and sequencing**

ChIP-seq and polyA-selected total RNA libraries were prepared with NEBNext® Ultra II DNA Library Prep Kit for Illumina. Optimum PCR cycles were determined via StepOnePlus qPCR using PowerTrack SYBR Green reagents (Thermo Scientific). Ribo-zero-selected total RNA libraries were prepared with NEBNext® Ultra II Directional DNA Library Prep Kit for Illumina.

ChIP-seq, 5′ dependent and independent small RNA, and polyA-selected total RNA libraries were sequenced on a HiSeq 1500 machine at the Gurdon Institute.

Total RNA was extracted using TRIzol reagent (Ambion, Life Technologies) and treated with Turbo DNase Kit (Invitrogen) according to the manufacturer's instructions. 1 µg total RNA was used for library preparation. Ribo-zero-selected total RNA libraries were sequenced on a HiSeq 2500 at the Cancer Institute, Cambridge. All sequencing was performed with SE50.

Short capped RNA sequencing from nucleoplasmic and chromatin gradients were sequenced at the LMS Sequencing Facility on a HiSeq 2500 with SE75.

**RNA extraction and real-time quantitative PCR of mature piRNAs**

Total RNA was extracted using TRIzol reagent (Ambion, Life Technologies) and treated with Turbo DNase Kit (Invitrogen) according to the manufacturer's instructions. 5 µg of total RNA was oxidized (25 mM $NaIO_4$/1X borate buffer) for 10 min at 25°C in the dark. 10 ng of total RNA was reverse-transcribed with a TaqMan Small RNA Assay Kit (Thermo Fisher) containing a gene-specific RT primer and a TaqMan MicroRNA Reverse Transcription Kit (Thermo Fisher), according to the manufacturer's instructions. Real-time

quantitative PCR was performed on 1 ml of RT reactions using TaqMan Universal Master Mix No AmpErase UNG (Life Technologies) on a OneStepPlus thermocycler (Thermo Fisher). All samples were run in triplicates. Expression levels were normalized to the reference gene U18.

### piRNA precursor length analysis

The analysis of piRNA precursor length was carried out as described in Beltran *et al* (2020), except for a bootstrap sample size of 5000 sequences owing to a greater number of detected piRNA precursor sequences across conditions. Degradation fragment analysis was carried out as described in Beltran *et al* (2020).

### RNA-sequencing data analysis

polyA and ribo-zero-selected total RNA-seq libraries were mapped using STAR v2.5.4b. to *C. elegans* ce11 genome carrying the piRNA sensor as an extra chromosome. STAR mapping parameters were "--outMultimapperOrder Random --outFilterMultimapNmax 5,000 --outFilterMismatchNmax 2 --winAnchorMultimapNmax 10,000 --alignIntronMax 1 --alignEndsType EndToEnd".

Adapter sequences were trimmed from 5′ dependent and 5′ independent small RNA libraries with cutadapt v 1.15 (Martin, 2011) using "-m 14 -M 32 -a TGGAATTCTCGGGTGCCAAGG" parameters and were mapped against ce11 genome with the piRNA sensor using STAR with ""--outMultimapperOrder Random --outFilterMultimapNmax 500 --outFilterMismatchNmax 1 --winAnchorMultimapNmax 10,000 --alignIntronMax 1 --alignEndsType EndToEnd" parameters. Aligned RNA-seq reads were sorted and indexed with samtools v1.6 (Li *et al*, 2009). Total RNA read counts against the annotations were generated by featureCounts v1.6 (Liao *et al*, 2014) with parameters "-T 12 -M –fraction -t exon" from a GTF file.

22 nt small RNA reads with a 5′ G bias were extracted from 5′ independent small RNA-sorted bam files with a custom PERL script. Read counts with an antisense orientation to the transcriptional direction of annotated genes were extracted by featureCounts v1.6 using "-T 12 -M --fraction -s 2 -t exon " parameters. We next calculated the antisense 22G small RNA density over 20 equal bins by TPM normalization for annotated genes. We then plotted corresponding mRNA TPM values in each bin and calculated the overlap between each bin with known HRDE-1, CSR-1 (Zhang *et al*, 2011), and piRNA targets (Bagijn *et al*, 2012). Antisense 22G small RNA profiles on the piRNA sensor and transposons were normalized as reads per million (RPM). A GTF file containing annotations for transposable elements was generated by RepeatMasker (Smit *et al*, 2015) v 4.1.0 run with rmblastn version 2.2.27+ using RepeatMasker database version 20140131, against the ce11 genome. Read counts of total and small RNA libraries on individual transposons were calculated using uniquely mapped reads. Normalized counts, variance-stabilized counts, log2 fold changes, and adjusted *P* values were obtained using DEseq2 v 1.18.1 (Love *et al*, 2014). For differential gene and transposon expression, candidates with a log2 fold change bigger than 1 or smaller than −1 as well as an adjusted *P* value smaller than 0.01 were considered as statistically significant differential expression. To overcome false-positive transposon differential expression, statistically significant upregulated transposon list was further

filtered with the ones that span or overlap with the exons of statistically significant upregulated protein-coding genes.

piRNA abundance was analyzed using 5′ dependent libraries. Small RNA read counts for piRNAs were obtained using featureCounts v1.6 with "-T 12 -M --fraction" parameter and normalized as reads per million (RPM). Cluster analysis of 5′-independent small RNA libraries showing the fold change in small RNAs mapped piRNA targets in the indicated mutants compared with wild type. Fold change is displayed in natural log.

To compare changes in small RNA targeting piRNA targets in between animal strains (Bagijn *et al*, 2012), small RNA reads per gene were counted and abundance calculated by correcting for library size using unique mapping reads (cutoff > 50 reads per million). The mean small RNA abundance per gene was calculated; next, the fold change was calculated by dividing the mean abundance in mutant animals by the mean abundance in wild-type animals.

Statistical significance for mature piRNA abundance and IP/MS was determined by two-sample *t*-test and Welch's *t*-test, respectively. To prevent zero values during log2 and log10 transformations, pseudo-count 1 was added for TPM and RPM normalizations.

### ChIP-sequencing analysis

ChIP-seq libraries were mapped to *C. elegans* ce11 genome with the piRNA sensor as an extra chromosome by BWA allowing multimappers. sam files were converted to bam format, and then, bam files were sorted and indexed using samtools v1.6. (Li *et al*, 2009) ChIP-seq peaks were generated using MACS2 v 1.4.3 (Zhang *et al*, 2008) peakcall function against input libraries. Common peaks between individual replicates with a *P* value smaller than 0.01 were considered as significant peaks. Linear fold enrichment was calculated with MACS2 bdgcmp function by disabling --SPMR option per replicate. Log2 and Z-score transformations were performed on RStudio with a custom script. All individual replicates were combined with RStudio Rtracklayer and Genomic Ranges packages. Seqplots was used to generate density plots and heat maps (Stempor & Ahringer, 2016). Class I, class II, and class III genes were classified by mean log2 fold enrichment of combined replicates over gene bodies of statistically significant upregulated genes. Briefly, we considered genes for which mutant log2 fold enrichment was 1.5 times greater than that of wild type, as class I. Similarly, genes with a mutant log2 fold enrichment 1.5 times smaller than that of wild type was considered as class III. The rest of the upregulated genes remaining between these ratios were considered as class II.

### Immunofluorescence stainings

Worms were picked on glass slides and dissected to extrude the germlines. Germlines were subsequently freeze-cracked on dry ice and fixed in cold 100% methanol for 20 min. Fixed slides were then washed in 0.05% Tween/PBS and incubated with diluted (1:1,000) anti-gfp (Abcam #ab290) and diluted (1:1,000) anti-mCherry antibodies (GeneTex #GTX128508) overnight. The following day, slides were washed in 0.05% Tween/PBS and incubated with diluted secondary antibodies for 1 h at room temperature and mounted in VECTASHIELD with DAPI. Representative images were taken on a

Leica SP8 confocal microscope with a 63× oil objective and 4× digital magnification. Single-plan images are shown.

### Image analysis—quantification of Integrator::gfp signal in prde-1::mCherry foci

To quantify Integrator-GFP signal within PRDE-1-mCherry foci, a minimum of 16 PRDE-1-mCherry foci from different nuclei were selected. Individual z-planes were chosen based on PRDE-1-mCherry maximum intensity. PRDE-1 foci were then segmented based on the threshold obtained by Otsu's method (Otsu, 1979). The resulting ROIs were used to measure the average intensity in the Integrator::gfp channel, either co-localizing with PRDE-1 or at a minimum of 10 random places elsewhere in the nucleus across the slide as a normalizing factor. The analysis was implemented with a custom script written in Python and using the scikit-image package.

### Materials availability

Transgenic animals will be submitted to the *C. elegans* Genetics Center.

## Data availability

The data sets and computer codes generated in this study are available in the following databases:

- RNA-seq (gene expression) Omnibus GSE149071 (https://www.ncbi.nlm.nih.gov/geo/query/acc.cgi?acc=GSE149071)
- ChIP-seq (genome-wide localization) Omnibus GSE149071 (https://www.ncbi.nlm.nih.gov/geo/query/acc.cgi?acc=GSE149071)
- Protein interaction IP-MS data PRIDE PXD018657 (https://www.ebi.ac.uk/pride/archive/projects/PXD018657)
- Bioinformatics computer scripts: GitHub (https://github.com/berkyurekac/ahmet_toolkit)

Expanded View for this article is available online.

## Acknowledgements
We thank Archana Yerra for critical reading of the manuscript, and Charles Bradshaw and Peter Williams for helping with data submission. We thank Andrea Frapporti for the help with image analysis and Alex Appert for the help with the ChIP protocol. We thank the Gurdon Institute Media Kitchen for their support in providing reagents and media. We thank Kay Harnish for his support in managing the Gurdon Institute Sequencing Facility. We thank Julie Ahringer's Lab for kindly sharing RNAi bacterial clones. We are grateful for the Miska Laboratory members, especially Kin Man Suen, for helpful discussions and advice, and Marc Ridyard for laboratory management and maintenance of our nematode collection. This work was supported by Cancer Research UK (C13474/A18583, C6946/A14492) and the Wellcome Trust (104640/Z/14/Z, 092096/Z/10/Z) to E.A.M; work in the Sarkies Lab was funded by the Medical Research Council (Transgenerational Epigenetic Inheritance and Evolution) and an EMBO Young Investigator Award (to P.S.). A.C.B. was supported by a Marie Skłodowska-Curie Individual Fellowship (747666) and an HFSP grant to E.A.M. (RPG0014/2015); G.F. was supported by an EMBO Long-Term fellowship (EMBO ALTF 1132-2018); and L.L. was supported by a Boehringer Ingelheim Fonds PhD fellowship. I.C.N. was supported by a Science without Borders Doctorate scholarship (205589/2014-6; CNPq, Brazilian Federal Government).

## Author contributions
EAM conceptualized the data; ACB, GF, LL, TB, EMW, EN, ICN, FBr, and AA investigated the data; ACB, TB, EN, JP, and FBr involved in formal analysis; GF wrote the original draft; EAM, ACB, LL, AA, PS, TB, and FBu wrote, reviewed, and edited the manuscript; LL visualized the data; EAM supervised the data; and EAM acquired funding.

## Conflict of interest

The authors declare no competing interests. E.A.M. is a founder and director of STORM Therapeutics Ltd. STORM Therapeutics had no role in the design of the study and collection, analysis, and interpretation of data as well as in writing the manuscript.

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
