## [Review Process File · The EMBO Journal]

The RNA Polymerase II subunit RPB-9 recruits the Integrator complex to terminate *C. elegans* piRNA transcription

Ahmet C. Berkyurek, Giulia Furlan, Lisa Lampersberger, Toni Beltran, Eva-Maria Weick, Emily Nischwitz, Isabela Navarro, Fabian Braukmann, Alper Akay, Jonathan Price, Falk Butter, Peter Sarkies, and Eric Miska

DOI: [10.15252/embj.2020105565](https://doi.org/10.15252/embj.2020105565)

Corresponding author(s): Eric Miska (eam29@cam.ac.uk)

Review Timeline:

Submission Date:	7th May 20
Editorial Decision:	9th Jun 20
Revision Received:	2nd Nov 20
Editorial Decision:	3rd Dec 20
Revision Received:	14th Dec 20
Accepted:	19th Dec 20

Editor: Stefanie Boehm

Transaction Report:

Thank you for submitting your manuscript for consideration by The EMBO Journal. Please also excuse the delay in communicating this decision to you, which was due to a delayed review process on account of the current pandemic. We have now however received three referee reports on your study, which are included below for your information.

As you will see, the reviewers overall express an interest in the study, but also raise several major concerns that would need to be addressed in a revised manuscript. In particular, the referees find that the interaction of RBP-9 with the Integrator complex should be assessed in further detail (ref#1 major point 1; ref#2 point 1(b)), including the functional role of this interaction (ref#3 major point 3). Additional experimental proof for the interaction/co-localization of RBP-9 with Integrator and functional analyses of Integrator in *rpb-9* mutants should thus be provided in the revised version. Furthermore, it will also be important to analyze the current data in further detail with respect to *rpb-9* dependence of piRNA targets, transposon regulation, as well as clarifying the finding that many upregulated genes do not exhibit a transcriptional upregulation (ref #1 major point 3; ref#2 points 1, 3, 4, 7, 9; ref#3 1, 2). Moreover, the referees note that the current manuscript would benefit from restructuring of the text, in particular the introduction, as well as additional clarifications, such as referee#1's point 2. In addition to addressing these key issues, please also carefully consider all other points the referees raise and revise the manuscript accordingly.

Referee #1:

Through an EMS screening, Berkuyrek et al. have identified RPB-9, a subunit of Pol II, as a factor required for the piRNA pathway in worms. In *rpb-9* mutants, some of the piRNA targets are de-silenced, mature piRNAs are decreased, and instead elongated piRNA precursors are accumulated. Accordingly, the authors propose that RPB-9 promotes Integrator-dependent cleavage of nascent piRNA precursors. This study is interesting and beneficial for the field, providing a link between the core Pol II subunit and piRNA biogenesis.

Major Points:

1. The authors propose that RPB-9 and TFIIIS cooperate to recruit Integrator to terminate the piRNA precursor transcription, but direct evidence is lacking. Do the authors detect the interaction of RPB-9 with TFIIIS and/or Integrator in their IP-MS data (Fig. 4E)? It has been reported that association between Pol II and TFIIIS is compromised in *rbp-9* mutant in yeast (Sigurdsson et al., Mol Cell 2010), so the observed effect in *rbp-9* mutant in worms might be due to the absence of TFIIIS. It would substantially strengthen the study if the authors can directly examine the recruitment of TFIIIS and/or Integrator to piRNA loci as well as the cooperative effect between RPB-9 and TFIIIS.
2. In page 19, the authors compare their own results (Fig. 6) with those in the co-submitted paper, but there is a leap in the logical flow here. The readers will not be able to understand what "~20 nt long 3' nascent RNA cleavage fragments" and "+38 (Fig. 6E)" etc. are. Much more careful explanation and additional analyses on the abundance and quality of piRNA precursors and mature piRNAs are required. In particular, the authors should discriminate the first and second pausing sites of Pol II and analyze the effect of RPB-9 on each type of piRNA precursors.
3. Throughout this manuscript, the authors' analyses on piRNA targets are arbitrary and subjective. For example, there are several other upregulated transposons in Fig. 3A, but no reason is provided for why the authors decided to focus on only two of those. Moreover, it is unclear if piRNA targeting is specifically enriched for Chapaev-2 and CEMUDR1 DNA compared to non-regulated transposons. In page 12, the authors state that "*rpb-9* and *prg-1* mutants share a total of 51 deregulated genes (22 upregulated and 29 downregulated) (Figure 3E), while *rpb-9* and *hrde-1* share 42 (36 upregulated and 6 downregulated) (Figure 3F)", but it is unclear if this degree of overlapping is significantly higher than that expected by chance. In Fig. 4G, it is unclear why the authors focused on T, not D or G. Fig. 4 requires not only up-regulated genes but also negative controls of non-regulated and down-regulated genes with clarification of their definitions. Rigorous statistical analyses with appropriate controls are required.

Minor Points:

1. In the title and abstract, the authors refer to "transcriptional elongation" and "high-fidelity transcription", but this study provides no direct evidence.
2. Page 11: No data is referenced for the statement "for both transposons, 22G siRNAs were present at high levels in wild-type animals but were decreased in *prg-1* mutants".
3. Page 11: Incomplete sentence "as is the ."
4. Page 13: "Figure S4B-D" should read "Figure S4D-F".
5. Page 13: "Figure S4E" and "Figure S4F" should read "Figure S4B" and "Figure S4C" respectively.
6. Page 15: "Figure 4G and 4F" should read "Figure 4G and 4H".
7. Fig. S1B: "C. elegans" and "H. sapiens" should probably be switched.
8. Figs. 3E, S3D etc.: Please indicate which base line data was used to calculate the "fold change".

Referee #2:

The manuscript by Berkyurek et al. describes the identification of the minor RNA Pol II subunit *rpb-9* as an important factor in piRNA biology in *C. elegans*. They do so through mapping of a genetic screen hit from an EMS screen on a piRNA-regulated GFP sensor. The authors explore the function of *rpb-9* in piRNA biology mainly through a series of NGS experiments comparing wildtype and *rpb-9* mutants.

The topic of specialized functions of basal gene expression machineries is rapidly developing and should interest a broad readership also outside the *C. elegans* piRNA field. The findings and supporting data are in general strong, but I have some reservations with parts of the analyses. In addition, I find the way the manuscript is structured confusing, blurring the key message. I do believe that these reservations can be addressed through extensive revision and re-analyses, but requiring little additional experimental work. If the manuscript is improved to resolve the below concerns, it will form an important contribution to the *C. elegans* piRNA field as well as to our understanding of RNA Pol II transcription in general.

Major concerns

1. With the current angle in the manuscript, the central message of RPB-9-dependence in piRNA expression seems rather predictable as piRNAs are known to be transcribed by RNAPII and RPB-9 has a known general role in RNAPII transcription. The title of the paper indicates a more interesting finding of a role of RPB-9 in transcriptional elongation specifically at piRNA loci. To focus the paper on the message of the title, I suggest the following revisions:
 - a. Move figures 5 and 6 up to be figure 3 and 4 so that the piRNA biogenesis phenotype gets a more central position and gets directly connected to the sensor-based identification. The TE deregulation in 3A-B could also be included here. Host gene reregulation effects could then be investigated at the end as more separate topic.
 - b. Address why *rpb-9* is important at some loci but not at others. Are there differences in promoter composition at piRNA loci that depend on *rpb-9* compared to those that are not (also within class I and II piRNA loci)? Why are lowly expressed genes more sensitive to *rpb-9* mutation? Does Integrator localize to *prg-1*-enriched foci in *rpb-9* mutants? (indicating a direct connection to the co-submitted paper)
2. Focus of the introduction. The introduction is rather long and includes several paragraphs which are not directly related to the topic of the paper. This makes it difficult for the reader to know which part to pay attention to and thus to understand the paper. I would suggest to trim the introduction considerably, focusing specifically on knowledge required to understand the question of 21U piRNA transcription. Additional introductory knowledge can then be added in compact format in the relevant context in the results or discussion section.
3. Given the >1500 deregulated host genes in *rpb-9* mutants, the authors should clarify if there are known piRNA pathway-related genes amongst these that could potentially explain the *rpb-9* phenotype as indirect. This is especially important for the 292 down-regulated genes.
4. The authors show that most upregulated genes (RNA level, Fig. 3C) do not show transcriptional upregulation and several are even transcriptionally downregulated (RNAPII ChIP, Fig. 4B-C). This suggests a likely scenario: The upregulation in *rpb-9* mutants is mostly post-transcriptional

(potentially piRNA-mediated) and *rpb-9* mutation causes a transcriptional down-regulation given its' general role in RNAPII transcription. The downregulation effect size (see concern #5) is however so small that it does not offset the post-transcriptional stabilization effect. This scenario should be addressed experimentally or through further analyses (the PCR analysis of unspliced vs spliced GFP mRNA in Fig 4F is too limited to draw a conclusion ruling out direct transcriptional defects in *rpb-9* mutants as indicated on page 14). In addition, it may also improve the analyses to focus on the germline-expressed genes as in S3D (see point 8).

Minor concerns

5. Z-score plotting. I find the z-score-based analyses of RNAPII ChIP-seq in Figure 4A-D very difficult to interpret. Z-scores show the number of standard deviations from the population mean, but it is not noted how the mean was calculated and one cannot deduce an effect size from these plots. In principle, the shown differences between control and *rpb-9* could represent a very small increase that would be unlikely to have biological relevance. Simply plotting log₂ fold-change in RNAPII association for the upregulated genes would be a simpler a more readable way to display the data.

6. Figure 3A: three dots (TEs) in the top of the panel seem even more upregulated that the three in focus in Figure 3B. What are these elements and why are they not mentioned in 3B and text?

7. Figure 3A: since piRNA pathway perturbation results in very different transposon de-regulation phenotypes between different model organisms and tissues, it would be helpful to state the expected outcome of deregulating 21U piRNA production - for example by relating to the phenotype of *prg-1* mutation.

8. Figure 3E-F: The refined analyses focusing on germline-expressed genes (Figure S3D) is much easier to interpret given the focus on germline genes than the analyses in 3E-F, which seem like an unnecessary detour. I find that this section could be strongly improved by omitting the current Figure 3D-F and replacing it with the germline-focused analyses in S3D.

9. Page 15: the authors refer to Fig 4G and write: "Interestingly, the mean expression levels of genes in bin T was higher in *rpb-9* mutants compared to wild type. This suggests that these genes are likely direct piRNA targets." The mean expression level is, however, also higher in *rpb-9* mutants for bins B to K, which have a similar expression level to bin T. How can the authors rule out that expression level rather than 22G siRNA density explains this *rpb-9* mutant phenotype for bin T?

10. Figure 4G-K: in which bin are the deregulated transposons and do they fit the authors' model?

11. Page 15: "indeed belong to this bin (Figure 4G and Figure 4F)." 4F should be 4H?

12. Page 12: "We observed a significant overlap..". The term 'significant' is better reserved for describing statistical test results. 'notable' instead?

Referee #3:

The piRNAs in *C. elegans* are 21-U RNAs, a population of 21-nt small RNAs characterized by a 1U bias and a characteristic sequence motif; 42 nt upstream of the start of the small RNA. 21-U RNAs appear to be derived from thousands of individual, autonomously expressed loci broadly scattered in two large clusters on chromosome IV.

In this manuscript, Berkyurek et al. show that a mutation of RNA polymerase II subunit RPB-9 can impact piRNA biogenesis and thereby piRNA-mediated regulation of gene expression in *C. elegans*. It was found that RPB-9 is required to promote the Integrator-dependent cleavage of 3' ends of nascent transcripts upon RNA Pol II backtracking for transcription termination at motif-dependent piRNA loci. Overall, this is an intriguing characterization of a new gene involved in the piRNA pathway in *C. elegans*. However, it would be difficult for the reader to comprehend the manuscript as is often the case for studies characterizing molecular pathways of small RNA biogenesis in *C. elegans*: At which step(s) of piRNA biogenesis is RPB-6 really functioning?

Major Criticisms

1. Among many perplexed results, results shown in Figure 4 are very confusing: Why and how did the majority of upregulated genes display unchanged or even reduced RNA pol II binding in *rpb9* animals, despite being upregulated (Class II and Class III genes)? Then the authors find a strong reduction in the amount of 22G siRNAs, which may explain why unchanged or reduced RNA pol II binding in *rpb9* animals still results in their upregulation. But how is RPB-9 required for the production of 22G siRNAs at a "subset" of piRNA targets? The authors state "class II (and to a minor extent class III) genes mostly resided in the last bin (T)." But piRNA sensor belongs to class III. Which of three classes do Chapaev-2 and CEMUDR1 belong to?
2. The authors find that transcriptional elongation at the piRNA sensor locus is in fact efficient in *rpb-9* animals (Figure 4), though the IP-MS data indicate that RPB-9 strongly interacts with components of the elongation machinery (Figure 4E). They also find that piRNA precursors are slightly longer than those observed in wild type (Figure 6). However, these precursors can still be cleaved in *rpb-9* animals. Thus it is hard to understand how the production of mature piRNAs is reduced but not abolished in *rpb-9* animals (Figure 5B). Figure 5C shows that levels of piRNA 21UR-1 from the piRNA sensor loci are not significantly reduced in *rpb-9* animals. Are piRNA precursor transcripts accumulated in *rpb-9* animals? Can the authors observe the precursors of 21UR-1 in the northern blots? Also how could such levels of 21UR-1 lead to the loss of 22 G siRNAs in *rpb-9* animals? Also how would the authors envisage that such an extremely low abundant piRNAs (Figure 3B and Figure 5D) can have a big impact on the regulation of gene expression of transposable elements such as Chapaev-2 and CEMUDR1?
3. Finally, the Integrator complex is known to be involved in 3' end formation of snRNAs (and probably some other classes of RNAs). Have the authors examined levels of snRNAs in *rpb-9* animals? Have the authors examined pre-mRNA splicing in *rpb-9* animals? Is the recruitment of the Integrator complex on 3' ends of piRNA precursors dependent on RPB-9? *mj261* mutation appears to interrupt translation within the TFIIIC domain. Is the TFIIIC domain the binding domain that interacts with the integrator complex?

Some other comments:

The text seems to be unnecessarily long, in particular Introduction should be shortened. The authors seem to have written the manuscript in a hurry. There are some errors in the text and Refs. For example, page 11 the third para: ----in the germline as is the . The citations of Bagijn et al 2012a and b appear the same paper. Refs are sloppy.

Referee #1:

Through an EMS screening, Berkyurek et al. have identified RPB-9, a subunit of Pol II, as a factor required for the piRNA pathway in worms. In *rpb-9* mutants, some of the piRNA targets are de-silenced, mature piRNAs are decreased, and instead elongated piRNA precursors are accumulated. Accordingly, the authors propose that RPB-9 promotes Integrator-dependent cleavage of nascent piRNA precursors. This study is interesting and beneficial for the field, providing a link between the core Pol II subunit and piRNA biogenesis.

Major Points:

1. The authors propose that RPB-9 and TFIIS cooperate to recruit Integrator to terminate the piRNA precursor transcription, but direct evidence is lacking. Do the authors detect the interaction of RPB-9 with TFIIS and/or Integrator in their IP-MS data (Fig. 4E)?

Reply:

During the course of our research, we looked for TFIIS or Integrator Complex proteins in the IP-MS data, but did not detect any of these proteins. This could be explained by: i-) A possible weak or transient interaction between RPB-9 and TFIIS or Integrator Complex that cannot be detected by conventional strategies. ii-) The interaction between RPB-9 and TFIIS or Integrator Complex might be mediated through RNA. iii-) The detection of interaction between RPB-9 and TFIIS or Integrator Complex might require very specific experimental conditions. E

*In order to explore the RPB-9/Integrator interaction further, , we also performed an INTS-6::GFP pull-down followed by RPB-9 western blot, using an anti-GFP antibody and a human anti-RPB-9 antibody. We added *rpb-9* RNAi control , and used high-yield *C. elegans* extracts with RNase inhibitors to protect RNA-mediated interactions. As you will see in the figure below, we detected a very weak band for RPB-9 compared to RNAi control, indicating a possible interaction between RPB-9 and INTS.*

It has been reported that association between Pol II and TFIIIS is compromised in *rbp-9* mutant in yeast (Sigurdsson et al., Mol Cell 2010), so the observed effect in *rbp-9* mutant in worms might be due to the absence of TFIIIS. It would substantially strengthen the study if the authors can directly examine the recruitment of TFIIIS and/or Integrator to piRNA loci as well as the cooperative effect between RPB-9 and TFIIIS.

Reply:

We agree that an additional experiment analyzing the recruitment of Integrator with RPB-9 will strengthen the manuscript considerably. For this, we carried out two major experiments: i-) Analysis of Integrator recruitment at piRNA loci in control and rbp-9 mutant animals via immunostaining

*We crossed our *rbp-9* mutants with animals in which the core Integrator subunit INTS-6 and the piRNA biogenesis factor PRDE-1 are endogenously tagged with GFP and mCherry respectively, and analyzed Integrator localization via immunostaining. In wild-type animals, the Integrator complex localizes throughout the nucleus, but accumulates specifically on piRNA genes, as indicated by the presence of INTS-6 clouds and their co-localization with PRDE-1 signals. In *rbp-9* mutants, Integrator is still homogeneously distributed in the nucleus, but the accumulation clouds are lost, and INTS-6/PRDE-1 co-localization is decreased. This indicates that RPB-9 is required to physically recruit the Integrator complex at piRNA loci, where its cleavage activity is necessary to terminate transcription of piRNA precursors. These results are now in Fig. 7F-G. We also amended the manuscript text with these new results, accordingly.*

ii-). Phenotypic analysis of *tfiis;rpb-9* double mutant animals for a cooperative effect between RPB-9 and TFIIS

A cross between *tfiis* and *rpb-9* mutants resulted in *tfiis;rpb-9* double mutants that failed to produce viable offspring (Appendix 3). F2 animals are fertile but lay fertilized embryos that die before hatching. This result suggests that *rpb-9* and *tfiis* are genetic interactors and may play a role in the same pathway. With this novel result, we amended the manuscript text.

INTS-6::GFP recruitment to piRNA loci by RPB-9 as detected by immuno-fluorescence together with IP-MS data as shown in Major Point#1 by Referee#1, we now provide strong evidence for a functional link between RPB-9 and TFIIS/Integrator on the piRNA loci. Due to space and text limitations in the manuscript, we have included only immuno-fluorescence and *tfiis;rpb-9* double mutant phenotypic analysis data in the revised version of the manuscript.

2. In page 19, the authors compare their own results (Fig. 6) with those in the co-submitted paper, but there is a leap in the logical flow here. The readers will not be able to understand what "~20 nt long 3' nascent RNA cleavage fragments" and "+38 (Fig. 6E)" etc. are. Much more careful explanation and additional analyses on the abundance and quality of piRNA precursors and mature piRNAs are required. In particular, the authors should discriminate the first and second pausing sites of Pol II and analyze the effect of RPB-9 on each type of piRNA precursors.

Reply:

We have now performed these analyses and included the results in Figure 7 and EV5. We show that the abundance of motif-dependent piRNA precursors is slightly decreased compared to wild type, while this is not the case for motif-independent precursors. We also clarified the text and added the information necessary to understand the comparison between our data and those of the co-submitted manuscript (Beltran et al.. 2020).

3. Throughout this manuscript, the authors' analyses on piRNA targets are arbitrary and subjective. For example, there are several other upregulated transposons in Fig. 3A, but no reason is provided for why the authors decided to focus on only two of those.

Reply:

We sincerely apologize for this confusion regarding the analysis of transposable elements. Throughout the manuscript, we focused on transposons that were statistically and significantly deregulated ($p_{adj} < 0.01$, $l2fc \geq 1$ analysis performed with DeSEQ2, RStudio) in both polyA-selected (Figure 3A) and Ribo-Zero depleted (Appendix 2A) RNA-seq data sets. With these criteria, only *Chapaev-2* and *CEMUDR1* appear to be misregulated. To eliminate this misunderstanding, we have now added extra explanation (red colour: $p_{adj} < 0.01$, $l2fc \geq 1$) on our figures

Moreover, it is unclear if piRNA targeting is specifically enriched for *Chapaev-2* and *CEMUDR1* DNA compared to non-regulated transposons.

Reply:

No, piRNA targeting is NOT enriched for DNA transposons over others. Previous publications from our group as well as from other laboratories have shown that piRNAs target all types of transposable elements in *C. elegans*. In the case of *rpb-9(mj261)* mutants, only *Chapaev-2* and *CEMUDR1* DNA transposons are upregulated. We have now modified the text to make this message more clear.

In page 12, the authors state that "rpb-9 and prg-1 mutants share a total of 51 deregulated genes (22 upregulated and 29 downregulated) (Figure 3E), while rpb-9 and hrde-1 share 42 (36 upregulated and 6 downregulated) (Figure 3F)", but it is unclear if this degree of overlapping is significantly higher than that expected by chance.

Reply:

We agree that the overlap of deregulated genes between the rpb-9 mutant and other piRNA pathway mutants is far from complete. We have observed this before for other components of the piRNA pathway (Akay A, Di Domenico T, Suen KM, Nabih A, Parada GE, Larance M, Medhi R, Berkyurek AC, Zhang X, Wedeles CJ, Rudolph KLM, Engelhardt J, Hemberg M, Ma P, Lamond AI, Claycomb JM, Miska EA. The Helicase Aquarius/EMB-4 Is Required to Overcome Intronic Barriers to Allow Nuclear RNAi Pathways to Heritably Silence Transcription. Dev Cell. 2017 Aug 7;42(3):241-255.e6. doi: 10.1016/j.devcel.2017.07.002. PMID: 28787591; PMCID: PMC5554785.). This is also true for piRNA pathway mutants in Drosophila.

However, to evaluate the significance of these overlaps and their linear regression coefficients, we used a hypergeometric test and representation factor (<http://nemates.org/MA/progs/representation.stats.html>), which is now included in the main text.

In Fig. 4G, it is unclear why the authors focused on T, not D or G.

Reply:

The genes in the other bins, although they are upregulated (mean), are not high responders to 22G levels, so they are potentially downstream or indirect targets, and not directly targeted by piRNA-dependent 22Gs.

Fig. 4 requires not only up-regulated genes but also negative controls of non-regulated and down-regulated genes with clarification of their definitions. Rigorous statistical analyses with appropriate controls are required.

Reply:

We agree. We have performed a Mann Whitney U statistical test on the Z-scores and plotted the mean Z-scores for corresponding transcription start sites (TSS) (+/-100 bp) in box plots. Our analysis shows that the differences between wild type and rpb-9 (mj261) mutants in RNA pol II enrichment on class I and class III genes are statistically significant ($p < 0.05$), whereas the differences on class II genes are not significant, confirming our conclusions. We also performed the same Pol II enrichment analysis on down-regulated genes and included the associated statistics (Mann Whitney U test). The majority of downregulated genes (94%) showed a decreased RNA pol II enrichment in rpb-9 (mj261) mutant, explaining the reduction in transcript levels. We have included these statistical tests on box plots next to the corresponding density plots of RNA pol II binding (Figure 4A-C and Figure EV2B-E). In addition to up and down regulated genes, we also checked RNA pol II binding on non regulated genes. RNA pol II enrichment in wild type and rpb-9 mutant conditions did not show notable differences.

Minor Points:

1. In the title and abstract, the authors refer to "transcriptional elongation" and "high-fidelity transcription", but this study provides no direct evidence.

Reply:

We have now modified the text to make it more clear.

2. Page 11: No data is referenced for the statement "for both transposons, 22G siRNAs were present at high levels in wild-type animals but were decreased in prg-1 mutants".

3. Page 11: Incomplete sentence "as is the ."

4. Page 13: "Figure S4B-D" should read "Figure S4D-F".

5. Page 13: "Figure S4E" and "Figure S4F" should read "Figure S4B" and "Figure S4C" respectively.

6. Page 15: "Figure 4G and 4F" should read "Figure 4G and 4H".

Reply:

We have updated the figure numbers, so that they are correctly referencing the text and displaying the additional results.

7. Fig. S1B: "C. elegans" and "H. sapiens" should probably be switched.

Reply: We decided to leave C. elegans at the top, as this is the model organism for our study.

8. Figs. 3E, S3D etc.: Please indicate which baseline data was used to calculate the "fold change".

Referee #2:

The manuscript by Berkyurek et al. describes the identification of the minor RNA Pol II subunit rpb-9 as an important factor in piRNA biology in *c. elegans*. They do so through mapping of a genetic screen hit from an EMS screen on a piRNA-regulated GFP sensor. The authors explore the function of rpb-9 in piRNA biology mainly through a series of NGS experiments comparing wildtype and rpb-9 mutants.

The topic of specialized functions of basal gene expression machineries is rapidly developing and should interest a broad readership also outside the *c. elegans* piRNA field. The findings and supporting data are in general strong, but I have some reservations with parts of the analyses. In addition, I find the way the manuscript is structured confusing, blurring the key message. I do believe that these reservations can be addressed through extensive revision and re-analyses, but requiring little additional experimental work. If the manuscript is improved to resolve the below concerns, it will form an important contribution to the *c. elegans* piRNA field as well as to our understanding of RNA Pol II transcription in general.

Major concerns

1. With the current angle in the manuscript, the central message of RPB-9-dependance in piRNA expression seems rather predictable as piRNAs are known to be transcribed by RNAPII and RPB-9 has a known general role in RNAPII transcription. The title of the paper indicates a more interesting finding of a role of RPB-9 in transcriptional elongation specifically at piRNA loci. To focus the paper on the message of the title, I suggest the following revisions:

a. Move figures 5 and 6 up to be figure 3 and 4 so that the piRNA biogenesis phenotype gets a more central position and gets directly connected to the sensor-based identification. The TE deregulation in 3A-B could also be included here. Host gene reregulation effects could then be investigated at the end as more separate topic.

Reply:

We agree with the reviewer that the story could be organized in a different way if more mechanistic aspects were available through crystallography or biophysics. However, with the available results, we decided to stick to the current version of the manuscript.

b. Address why rpb-9 is important at some loci but not at others. Are there differences in promotor composition at piRNA loci that depend on rpb-9 compared to those that are not (also within class I and II piRNA loci)?

Reply:

Motif-independent piRNAs are very lowly expressed and difficult to capture with the standard small RNA library preparation protocols. Therefore, we are not able to make a strong conclusion for motif-independent piRNAs in our manuscript.

Why are lowly expressed genes more sensitive to rpb-9 mutation?

Reply:

Our genome-wide analysis shows a wide range of deregulated genes in rpb-9 mutants. We specifically direct our attention to certain classes of genes, according to their tendency of misregulation (upregulated vs downregulated genes) and to their PolIII binding profiles. However, we never draw any conclusion on their expression level and their "sensitivity" to

rpb-9 mutation. We have no evidence for lowly-expressed genes being more impacted by the *rpb-9* mutation.

Does Integrator localize to *prg-1*-enriched foci in *rpb-9* mutants? (indicating a direct connection to the co-submitted paper)

Reply:

We agree with the reviewer that additional evidence is required regarding the functional link between RPB-9 and Integrator, as also suggested by Refere#1 in Major Point 1, paragraph 2. As explained above, we performed additional experiments and now display these new results in Figure 7F-G.

2. Focus of the introduction. The introduction is rather long and includes several paragraphs which are not directly related to the topic of the paper. This makes it difficult for the reader to know which part to pay attention to and thus to understand the paper. I would suggest to trim the introduction considerably, focusing specifically on knowledge required to understand the question of 21U piRNA transcription. Additional introductory knowledge can then be added in compact format in the relevant context in the results or discussion section.

Reply:

We agree with the reviewer's comments. We have shortened and amended the introduction.

3. Given the >1500 deregulated host genes in *rpb-9* mutants, the authors should clarify if there are known piRNA pathway-related genes amongst these that could potentially explain the *rpb-9* phenotype as indirect. This is especially important for the 292 down-regulated genes.

Reply:

*We have checked the expression levels of all well characterised piRNA and nuclear RNAi pathway genes: *prg-1*, *hrde-1*, *prde-1*, *tofu-5*, *tofu-4*, *snp-4*, *set-32*, *pid-1*, *desp-1*, *mut-16*, *mut-7* and *rrf-1*. We have NOT observed any of these genes among the statistically significantly deregulated ones in *rpb-9(mj261)* RNA-seq (both polyA selected and Ribo-Zero depleted). This suggests that the phenotype we observe is a direct effect of the role of *rpb-9* in transcription termination at piRNA genes, rather than an indirect effect of a lower expression of piRNA pathway genes.*

4. The authors show that most upregulated genes (RNA level, Fig. 3C) do not show transcriptional upregulation and several are even transcriptionally downregulated (RNAPII ChIP, Fig. 4B-C).

Reply:

Our data shows that transcriptionally upregulated genes show differential patterns of RNA pol II binding on the loci. Please note that transcriptional upregulation and polymerase binding are different concepts. We have amended the text to make this message more clear for the readers.

This suggests a likely scenario: The upregulation in *rpb-9* mutants is mostly post-transcriptional (potentially piRNA-mediated) and *rpb-9* mutation causes a transcriptional down-regulation given its general role in RNAPII transcription. The downregulation effect size (see concern #5) is however so small that it does not offset the post-transcriptional stabilization effect. This scenario should be addressed experimentally or through further analyses (the PCR analysis of unspliced vs spliced GFP mRNA in Fig 4F is too limited to draw a conclusion ruling out direct transcriptional defects in *rpb-9* mutants as indicated on

page 14). In addition, it may also improve the analyses to focus on the germline-expressed genes as in S3D (see point 8).

Reply:

While we cannot exclude that rpb-9 affects the piRNA sensor and Class II and Class III genes also directly (by transcribing these loci), we believe that the main component of the desilencing phenotype is provided by the piRNA-mediated post-transcriptional regulation.

The line of thought undertaken by the referee does not take into account that the starting scenario is one in which piRNA-mediated silencing is efficient: the animals on which we performed the screen are robustly silencing the piRNA sensor (and presumably all their other targets) prior to EMS treatment.

If rpb-9 had a prominent direct role in transcription at the piRNA sensor, we would expect a decrease in transcription at the sensor from already very low levels to levels that would probably not be enough to generate and sustain a robust 22G silencing response. This would result in a drop of (tertiary) 22G levels, which would prevent efficient co-transcriptional silencing. Transcription would then be allowed to continue, and the accumulation of spliced transcripts (in presence of wt levels of mature piRNAs) would eventually be enough to switch on the 22G response once again. In this scenario, the levels of GFP (nascent) transcript would fluctuate but would never be allowed to increase past the threshold needed to trigger co-transcriptional silencing, and the animals would never produce a functional GFP protein.

Since our screen is based on the ability of detecting sensor desilencing by looking at GFP protein expression, a role for rpb-9 exclusively in co-transcriptional silencing (i.e. by merely promoting sensor transcription) is not compatible with our screen - we would have not found rpb-9 alleles in this situation. In agreement with this, in our screen we could not retrieve any alleles of other core polymerase subunits.

On the other hand, a scenario in which piRNA levels are reduced as a primary consequence of the rpb-9 mutation indeed results in the inability of the animal to post-transcriptionally silence (spliced) GFP transcripts, which can then be translated into functional proteins. The decrease in piRNA levels would also result in a depletion of 22G siRNAs, which would fail to provide co-transcriptional silencing. As a result, GFP proteins would be stably expressed. This is exactly what we see in our mutants. For these reasons, we do not think that it will be necessary to perform additional experiments or analyses to clarify this point.

Regarding the analysis focused on germline expressed genes, we believe that this would result in loss of information. Filtering the whole dataset by germline-specific genes would hide events of transcriptional upregulation of somatic genes in the germline.

Minor concerns

5. Z-score plotting. I find the z-score-based analyses of RNAPII ChIP-seq in Figure 4A-D very difficult to interpret. Z-scores show the number of standard deviations from the population mean, but it is not noted how the mean was calculated and one cannot deduce an effect size from these plots. In principle, the shown differences between control and rpb-9 could represent a very small increase that would be unlikely to have biological relevance. Simply plotting log₂ fold-change in RNAPII association for the upregulated genes would be a simpler a more readable way to display the data.

Reply:

We always use Z-scores to compare ChIP-seq data on different sets of genes in our publications as Z-scores show how many standard deviations below or above the population mean a raw score is. For this reason, we decided to stick to Z-score graphs in the revised manuscript. At the same time, we are showing here an example of log₂ Fold Enrichment graphs for upregulated Class I, II and III genes, which show the same patterns as presented in the manuscript. The same is true for all other similar graphs (EV figures).

6. Figure 3A: three dots (TEs) in the top of the panel seem even more upregulated than the three in focus in Figure 3B. What are these elements and why are they not mentioned in 3B and text?

Reply:

*Throughout the manuscript, we focused on transposons that were significantly deregulated ($p_{adj} < 0.01$, $|2fc| \geq 1$, analysis performed with DeSEQ2, RStudio) in both polyA-selected (Figure 3A) and Ribo-Zero depleted (Appendix 2A) RNA-seq data sets. With these criteria, only *Chapaev-2* and *CEMUDR1* appear to be misregulated. To eliminate this misunderstanding, we have now added an extra explanation (red colour: $p_{adj} < 0.01$, $|2fc| \geq 1$) on our figures.*

7. Figure 3A: since piRNA pathway perturbation results in very different transposon deregulation phenotypes between different model organisms and tissues, it would be helpful to state the expected outcome of deregulating 21U piRNA production - for example by relating to the phenotype of *prg-1* mutation.

Reply:

*We have provided the comparison between *rpb-9* and *prg-1* transcriptomes in Figure 3E of our original manuscript. We have now also added the corresponding statistical tests to assess the significance of the overlap of misregulated genes between the two mutants.*

8. Figure 3E-F: The refined analyses focusing on germline-expressed genes (Figure S3D) is much easier to interpret given the focus on germline genes than the analyses in 3E-F, which seem like an unnecessary detour. I find that this section could be strongly improved by omitting the current Figure 3D-F and replacing it with the germline-focused analyses in S3D.

Reply:

We agree with the reviewer that a germline-focused analysis is the ideal analysis. However, our datasets were obtained from whole-animal samples, and not from germline-dissected ones. While we could in theory replace the panels in the main figure with S3D and draw the same conclusions, this would result in loss of information regarding the piRNA sensor, which is not present in the animals used for the germline-dissected analysis originally presented in S3D. Therefore, we added S3D to the main figure as panel 3G, to show the whole-animal and germline-dissected data together. To show that the overlaps between different piRNA pathway mutants are higher than that expected by chance, we now show the significance levels with hypergeometric test and representation factors.

9. Page 15: the authors refer to Fig 4G and write: "Interestingly, the mean expression levels of genes in bin T was higher in *rpb-9* mutants compared to wild type. This suggests that these genes are likely direct piRNA targets." The mean expression level is, however, also higher in *rpb-9* mutants for bins B to K, which have a similar expression level to bin T. How can the authors rule out that expression level rather than 22G siRNA density explains this *rpb-9* mutant phenotype for bin T?

Reply:

The genes in the other bins, although they are upregulated (mean), are not high responders to 22G levels. We believe they are potentially downstream or indirect targets, and not directly targeted by piRNA-dependent 22Gs.

10. Figure 4G-K: in which bin are the deregulated transposons and do they fit the authors' model?

Reply:

We agree with the reviewer that a transposon-focused analysis with small RNA/total RNA binning is vital to strengthen the manuscript. For this, we have provided new data where we divided all transposable elements into 10 equal bins, each containing 764 transposons. Then, we plotted the 22G siRNA density in an increasing order with corresponding transposon expression levels. We observed that Chapaev-2 and CEMUDR1 DNA transposons reside in the last bin, which has the highest 22G siRNA density, just like the piRNA sensor resides in the last bin of the mRNA analysis. This result is now in Figure 5E. With this new data, we have updated the manuscript text.

Protein coding genes and transposable elements analysis were performed separately due to two reasons: i-) Annotations for protein coding genes and transposons come from different sources. ii-) We used different settings with the bioinformatics pipelines for the analysis of protein coding genes and transposons. One important difference is using multi mapping reads in protein coding genes versus unique mapping reads in transposable elements.

11. Page 15: "indeed belong to this bin (Figure 4G and Figure 4F)." 4F should be 4H?

Reply:

We apologize for this mistake. We updated the manuscript text with the correct figure numbers.

12. Page 12: "We observed a significant overlap..". The term 'significant' is better reserved for describing statistical test results. 'notable' instead?

Reply:

We have now included an appropriate statistical test to show significance (main text page 12, describing figures 3E-G).

Referee #3:

The piRNAs in *C. elegans* are 21-U RNAs, a population of 21-nt small RNAs characterized by a 1U bias and a characteristic sequence motif; 42 nt upstream of the start of the small RNA. 21-U RNAs appear to be derived from thousands of individual, autonomously expressed loci broadly scattered in two large clusters on chromosome IV.

In this manuscript, Berkuyrek et al. show that a mutation of RNA polymerase II subunit RPB-9 can impact piRNA biogenesis and thereby piRNA-mediated regulation of gene expression in *C. elegans*. It was found that RPB-9 is required to promote the Integrator-dependent cleavage of 3' ends of nascent transcripts upon RNA Pol II backtracking for transcription termination at motif-dependent piRNA loci. Overall, this is an intriguing characterization of a new gene involved in the piRNA pathway in *C. elegans*. However, it would be difficult for the reader to comprehend the manuscript as is often the case for studies characterizing molecular pathways of small RNA biogenesis in *C. elegans*: At which step(s) of piRNA biogenesis is RPB-6 really functioning?

Major Criticisms

1. Among many perplexed results, results shown in Figure 4 are very confusing: Why and how did the majority of upregulated genes display unchanged or even reduced RNA pol II binding in *rpb9* animals, despite being upregulated (Class II and Class III genes)?

Reply:

*Figure 4 simply presents observations. The mechanistic explanation lies in the following sections of the paper. According to our model, Class II and Class III genes are upregulated in *rpb-9* mutants as a result of a defect in piRNA-mediated silencing. Rpb-9 is required to produce sufficient amounts of mature piRNAs, which induce post-transcriptional target silencing and 22G-mediated co-transcriptional silencing. This is independent from RNA PolIII binding, hence the apparent discrepancy between expression and Pol II enrichment.*

Then the authors find a strong reduction in the amount of 22G siRNAs, which may explain why unchanged or reduced RNA pol II binding in *rpb9* animals still results in their upregulation. But how is RPB-9 required for the production of 22G siRNAs at a "subset" of piRNA targets?

Reply:

We have responded to this in the revised discussion (second paragraph of the discussion, page 23).

The authors state "class II (and to a minor extent class III) genes mostly resided in the last bin (T)." But piRNA sensor belongs to class III.

Reply:

Although not all class III genes reside in the last bin (which, for the reasons explained in the main text (main text page 16) is the one containing high-responders to 22G-mediated silencing), the piRNA sensor clearly does. This confirms that it is a target of piRNA-mediated silencing, and suggests that the other mRNAs present in this bin are as well.

Which of three classes do Chapaev-2 and CEMUDR1 belong to?

Reply:

These transposons belong to Class III (upregulated, with decreased Pol II binding). Pol II binding profiles over these transposons are now shown in Figure 4E. We modified the manuscript text accordingly.

2. The authors find that transcriptional elongation at the piRNA sensor locus is in fact efficient in *rpb-9* animals (Figure 4), though the IP-MS data indicate that RPB-9 strongly interacts with components of the elongation machinery (Figure 4E).

Reply:

*Yes, probably elongation at the piRNA sensor locus does not require *rpb-9*. If it did, the sensor would be silenced or not strongly derepressed. We believe that the effect observed in the *rpb-9* mutant on the sensor ultimately depends on the piRNAs that are upstream of 22G siRNAs.*

They also find that piRNA precursors are slightly longer than those observed in wild type (Figure 6). However, these precursors can still be cleaved in *rpb-9* animals. Thus it is hard to

understand how the production of mature piRNAs is reduced but not abolished in rpb-9 animals (Figure 5B).

Reply:

We speculate that longer piRNA precursors somehow pose a problem to the 3' end processing machinery that matures precursors into functional piRNAs. We do not have, however, the necessary data to support this hypothesis. We believe that exploring this question will necessitate numerous additional experiments, which are beyond the scope of this paper.

Figure 5C shows that levels of piRNA 21UR-1 from the piRNA sensor loci are not significantly reduced in rpb-9 animals. Are piRNA precursor transcripts accumulated in rpb-9 animals?

Reply:

Unfortunately, our datasets don't allow us to explore piRNA precursor abundance at the single-locus level. Additionally, piRNA precursors are expressed at such low levels that even Northern Blot detection is almost impossible. We therefore present only aggregated data showing the mean abundance (Figure EV5) and the mean length (Figure 7A-D) of all piRNA precursors, according to type (motif-dependent and motif-independent) and fraction (nascent or nucleoplasmic). We cannot make conclusions regarding the precursor of piRNA 21UR-1.

Can the authors observe the precursors of 21UR-1 in the northern blots?

Reply:

piRNA precursors are found in very low abundance and previous work from our laboratory proved them to be very problematic to be detected via radioactive Northern Blots. For this reason, we were not able to perform successful Northern Blots for these specific targets.

Also how could such levels of 21UR-1 lead to the loss of 22 G siRNAs in rpb-9 animals?

Reply:

We speculate that there exists a specific threshold for piRNAs to achieve efficient silencing. We already mention this in our discussion section (second paragraph of discussion, page 22).

Also how would the authors envisage that such an extremely low abundant piRNAs (Figure 3B and Figure 5D) can have a big impact on the regulation of gene expression of transposable elements such as Chapaev-2 and CEMUDR1?

Reply:

We speculate piRNA thresholds and 22G amplification loops can have a big impact on the regulation.

3. Finally, the Integrator complex is known to be involved in 3' end formation of snRNAs (and probably some other classes of RNAs). Have the authors examined levels of snRNAs in rpb-9 animals?

Reply:

Yes, we have analyzed 129 annotated snRNA genes from ENSEMBL and found that 12 of them were significantly down-regulated. In addition to piRNAs, RPB-9 might have a role for some snRNA gene transcription.

We have analysed the levels of the 129 annotated snRNAs according to ENSEMBL and found that the majority of them (117) was not downregulated. We did observe a slight downregulation for the remaining 12, but we could not assess if these snRNA transcripts displayed a termination defect.

Have the authors examined pre-mRNA splicing in rpb-9 animals?

Reply:

We don't think that this analysis would be biologically relevant for the conclusions of the paper, especially since piRNAs, which are the main subject of our investigation, do not present the exon/intron structure typical of mRNAs. Additionally, we would require nascent-RNA libraries for this analysis.

Is the recruitment of the Integrator complex on 3' ends of piRNA precursors dependent on RPB-9?

Reply:

We thank the reviewer for this important suggestion. Using immuno-fluorescence experiments, we now show that RPB-9 recruits the Integrator complex to piRNA loci (Figure 7F-G).

mj261 mutation appears to interrupt translation within the TFIIC domain. Is the TFIIC domain the binding domain that interacts with the integrator complex?

Reply:

Unfortunately, we cannot answer this question with the available data. This would require additional biochemical experiments, as well as the generation of viable truncation mutants in which to examine Integrator localization. We believe that this could be a subject for a follow-up study but is certainly beyond the scope of this manuscript.

Some other comments:

The text seems to be unnecessarily long, in particular Introduction should be shortened. The authors seem to have written the manuscript in a hurry. There are some errors in the text and Refs. For example, page 11 the third para: ---in the germline as is the . The citations of Bagijn et al 2012a and b appear the same paper. Refs are sloppy.

Reply:

We have modified the manuscript text and made the necessary corrections in the references.

Thank you for submitting your revised manuscript. Please see below for the comments of the three initial referees on the revised version. The reviewers overall find that their comments have been addressed, but referee #1 and referee #3 raise remaining issues that should be resolved in a final round of revision. Please revise the manuscript accordingly and add to the discussion where applicable. As referee #1 also notes, please provide the custom code/scripts (by submitting them as Computer Code EV files or making them accessible in a public repository). Please also provide a brief point-by-point response to the comments, when submitting the revised version. In addition, I would also like to ask you to address a number of editorial issues that are listed in detail below. Please make any changes to the manuscript text in the attached document only using the "track changes" option. Once the remaining issues are resolved, we will be happy to formally accept the manuscript for publication.

REFEREE REPORTS

Referee #1:

The authors have adequately addressed most of previous concerns. The following points should be considered before publication.

1. Although the authors have added explanation about Integrator in page 20, it might be still difficult for the readers to understand the relationship between the 38-nt peak in Fig. 7H and the ~20 nt long 3' nascent RNA cleavage fragments. The authors should make it clear that the ~38 nt 21U-RNA precursors are produced by Integrator-mediated cleavage (and consequently the ~20 nt long 3' nascent RNA cleavage fragments).
2. The authors should clarify (either in the figure itself or in the legend) that the model in Fig. 8 applies to the motif-dependent piRNA pathway.
3. Citations (e.g., "(Izban & Luse, 1992), and others)" in page 5) and references (e.g., "[PREPRINT]") are incomplete.
4. Fig. S1B: "C. elegans" and "H. sapiens" should probably be switched (double-check if the amino-acid sequences and their labels are matching).
5. All the custom bioinformatic scripts should be made publicly available.

Referee #2:

The authors have revised the manuscript to address the major concerns and provided sufficient clarifying information. I therefore recommend accepting the revised manuscript for publication.

Referee #3:

The paper has been improved and the authors have addressed most of the reviewers' concerns. Studies of this sort are good references and resources for further comparisons. However, I still think that if this manuscript is to rise to the level required for publication in EMBO J, the authors should examine pre-mRNA splicing in *rpb-9* animals. This is because the authors found that 12 of 129 annotated snRNA genes were significantly down-regulated in *rpb-9* animals, indicating that RPB-9 may well have a role for some snRNA gene transcription, which in turn may have an impact on splicing, thereby affecting many genes including genes involved in piRNA biogenesis.

We thank the referees for their comments and suggestions, to which we have replied below:

Referee #1:

The authors have adequately addressed most of previous concerns. The following points should be considered before publication.

1. Although the authors have added explanation about Integrator in page 20, it might be still difficult for the readers to understand the relationship between the 38-nt peak in Fig. 7H and the ~20 nt long 3' nascent RNA cleavage fragments. The authors should make it clear that the ~38 nt 21U-RNA precursors are produced by Integrator-mediated cleavage (and consequently the ~20 nt long 3' nascent RNA cleavage fragments).
2. The authors should clarify (either in the figure itself or in the legend) that the model in Fig. 8 applies to the motif-dependent piRNA pathway.
3. Citations (e.g., "(Izban & Luse, 1992), and others)" in page 5) and references (e.g., "[PREPRINT]") are incomplete.
4. Fig. S1B: "C. elegans" and "H. sapiens" should probably be switched (double-check if the amino-acid sequences and their labels are matching).
5. All the custom bioinformatic scripts should be made publicly available.

Reply:

1. We have now clarified the paragraph relative to Figure 7H.
2. We added this information in the figure legend (Figure 8).
3. Citations have been corrected.
4. We changed this so that the labels match the amino-acid sequences.
5. We made sure that all the scripts are available and added the corresponding links to the Data Availability section.

Referee #2:

The authors have revised the manuscript to address the major concerns and provided sufficient clarifying information. I therefore recommend accepting the revised manuscript for publication.

Referee #3:

The paper has been improved and the authors have addressed most of the reviewers' concerns. Studies of this sort are good references and resources for further comparisons. However, I still think that if this manuscript is to rise to the level required for publication in EMBO J, the authors should examine pre-mRNA splicing in *rpb-9* animals. This is because the authors found that 12 of 129 annotated snRNA genes were significantly down-regulated in *rpb-9* animals, indicating that RPB-9 may well have a role for some snRNA gene transcription, which in turn may have an impact on splicing, thereby affecting many genes including genes involved in piRNA biogenesis.

Reply:

We agree that we cannot formally exclude the presence of a splicing defect affecting mRNAs encoding for piRNA pathway components in *rpb-9* mutants. This could indeed lead to poor protein synthesis and indirectly impact the silencing status of piRNA targets. However, we believe that the desilencing phenotype we observe at piRNA targets in *rpb-9* mutants is mostly direct and due to the defect in piRNA biogenesis: piRNA genes do not show the canonical exon/intron structure of coding genes and are not spliced. Hence, they can be nothing but direct targets of RPB-9/Integrator activity. Since piRNAs are the major primary signal that is required for initiation of silencing, it is natural to conclude that a great proportion of this desilencing phenotype is directly due to the action of RPB-9 and Integrator at piRNAs.

Thank you again for submitting the final revised version of your manuscript. I am pleased to inform you that we have now accepted it for publication in The EMBO Journal.

Corresponding Author Name: Eric A Miska

Manuscript Number: EMBOJ-2020-105565R